# Molecular dissection of colorectal cancer in pre-clinical models identifies biomarkers predicting sensitivity to EGFR inhibitors

Moritz Schütte et al.#

Colorectal carcinoma represents a heterogeneous entity, with only a fraction of the tumours responding to available therapies, requiring a better molecular understanding of the disease in precision oncology. To address this challenge, the OncoTrack consortium recruited 106 CRC patients (stages I–IV) and developed a pre-clinical platform generating a compendium of drug sensitivity data totalling >4,000 assays testing 16 clinical drugs on patient-derived *in vivo* and *in vitro* models. This large biobank of 106 tumours, 35 organoids and 59 xenografts, with extensive omics data comparing donor tumours and derived models provides a resource for advancing our understanding of CRC. Models recapitulate many of the genetic and transcriptomic features of the donors, but defined less complex molecular sub-groups because of the loss of human stroma. Linking molecular profiles with drug sensitivity patterns identifies novel biomarkers, including a signature outperforming *RAS/RAF* mutations in predicting sensitivity to the EGFR inhibitor cetuximab.

#A full list of authors and their affiliations appears at the end of the paper.

Colorectal cancer (CRC) is a clinically challenging, heterogeneous, disease representing the third most frequent cancer worldwide. CRCs can be classified within distinct molecular groups, although the clinical utility of this classification has not been demonstrated so far[1–5]. Only a fraction of advanced CRCs respond to the chemotherapeutic agents 5-fluorouracil (5-FU), irinotecan or oxaliplatin. Antibodies targeting the epidermal growth factor receptor (EGFR) offer therapeutic options, but have failed in the adjuvant setting[6]. BRAF, KRAS and NRAS (ref. 7) mutations are routinely used as predictive markers of resistance to the EGFR blockade. However a significant fraction of wild-type tumours remain unresponsive to cetuximab targeting EGFR (refs 8,9) thus requiring novel biomarkers predicting treatment outcomes.

Several pre-clinical studies based on in vivo or in vitro models of CRC have been reported[10–16], but without investigating their complex molecular landscapes nor comparing directly the different model systems.

Here we report an integrative pre-clinical approach based on the establishment and extensive molecular characterization of a large CRC biobank consisting of organoids and xenografts derived from a cohort of 106 patients representative of all CRC subtypes. Analysis of the responses of in vitro and in vivo models to a panel of clinically relevant therapeutic agents identifies gene signatures associated with objective drug response patterns.

## Results

**Establishment of the OncoTrack CRC pre-clinical platform**. The workflow of the OncoTrack (OT) study is summarized in Fig. 1. We collected from a prospective CRC cohort of 106 patients a total of 116 resected tissue samples with matched blood samples, comprising 89 primary tumours (ranging from stage I to IV) and 27 metastases as donors for generating a biobank of pre-clinical experimental models. We established in vitro and in vivo models with a success rate of approximately 60% in both systems. BRAF-mutated tumours engrafted with a higher efficiency (10/11 cases Fisher's exact test P = 0.04), likely reflecting their aggressive behaviour[17]. Here, we report the analysis of 46 patient-derived organoid cultures (PDO) and 59 xenografts (PDX) (Supplementary Data 1). Nineteen tumours were modelled in both systems, out of which five PDOs were derived from a PDX and two PDXs were established from a PDO. The topological expression of selected CRC markers was similar between models and their matched donor specimens (Supplementary Fig. 1a). PDO cultures formed organized structures featuring a cell-free lumen and proliferating KI67-positive cells, maintained after transfer into 384-well microtiter plates (Supplementary Fig. 1b; Supplementary Movie 1). We sequenced the genomes, exomes and transcriptomes of the donor cohort and of their matched untreated models, and established a drug-screening platform testing mechanistic compounds and chemotherapeutics, used in clinical standard of care.

**Molecular landscapes of the OT tumours and derived models**. We compared the genomic and transcriptome landscapes of the OT tumours with their derived pre-clinical models by integrating whole genome (WGS), whole exome (WES) and RNA sequencing data. We inferred the tumour purity from WGS data (Supplementary Data 1) and excluded samples with < 20% tumour content for the mutation scoring (final n = 101 samples/96 patients) (Methods and Supplementary Data 2–4). We called copy number variants (CNVs) and somatic mutations using matched patient germline DNAs as reference. Relevant mutations were sieved based on their expression and predicted damaging effects (methods and Supplementary Data 3). Microsatellite-

instable (MSI) samples were near-diploid, whereas microsatellite-stable (MSS) samples were either hyperploid (40 cases) or hypoploid (43 cases) with pervasive loss of heterozygosity (LOH) (Supplementary Fig. 2a). Deletions and focal amplifications were maintained in the models (Supplementary Data 3–5), while chromosomal instability was further accentuated (Supplementary Fig. 2b). We detected a total of 145 gene fusions, including the known driver event PTPRK-RSPO3 (ref. 18) (Supplementary Data 4). Novel fusions impacting CRC-relevant pathways inactivated APC or SMAD4, or were activating fusions such as a TRIM24-BRAF (in 196_T MSI) (Fig. 2a) predicted to trigger the conformational activation of the BRAF serine/threonine kinase domain, as observed in pilocytic astrocytoma[19] and melanoma[20]. FDFT1-FZD3 (Fig. 2a) and truncating fusions in the negative regulators of Wnt ZNRF3 and DACH1, were predicted to activate the Wnt pathway. Recurrent fusions truncated the haploinsufficient chromatin organizer CTCF and the solute carrier SLC12A2 (NKCC1) regulating the Cl⁻ flux in the intestinal crypt, but their contribution to CRC pathogenesis remain to be demonstrated. One xenograft harboured an ALK fusion as sole driver event (Supplementary Data 4), however we lacked the corresponding patient tumour.

The mutational profiles of the OT cohort and of the TCGA study[21] were very similar (Fig. 2b), demonstrating that our cohort represented the breadth of the CRC genetic landscape and that metastatic tumours did not show a biased mutation pattern. Nonetheless, the OT cohort displayed higher frequency of mutations in SOX9, maintaining the intestinal cell progenitors pool, (13 versus 4% in TCGA), and in TP53 (71 versus 51% in TCGA) (Supplementary Fig. 3a,b) (Fisher's exact test, Benjamini–Hochberg (BH) adjusted P = 0.04 and 0.02, respectively) overrepresented in stages III and IV (Fisher's exact test, P = 0.012) and often homozygous in hyperploid tumours (Student's t-test, P = 0.0008).

The genetic profiles of the models were generally concordant with their matched donor tumours. However, a number of models displayed clonal and sub-clonal differences irrespective of the tumour stage, with cases of extreme divergences such as for 118_T1 and 165_T and the sibling models derived from 227_T (Fig. 2c; Supplementary Data 6). These differences were more likely reflecting the intra-tumour heterogeneity (ITH) with different mutations found in distant regions of the tumour[22] sampled during model establishment than a genetic drift after serial passages. Early and late passage cell cultures showed virtually identical mutation patterns (Supplementary Fig. 4a,b), as well as five PDXs derived from tumour 150_MET1 (Supplementary Data 7), supporting this hypothesis. Along those lines, a comparable molecular analysis of breast cancer models showed only minimal clonal selection after serial transplantations[23]. Interestingly, we observed no discordances for driver mutations in BRAF and KRAS, although 118_MET and 227_T donor had KRAS homozygous mutations, whereas their respective models were heterozygous, reflecting ITH for CNVs. Clonality analysis with SciClone[24] identified mutation clusters private to either patient or model, or to one of the sibling models (Fig. 3a, Supplementary Fig. 5 and Supplementary Data 8). Only 3% of the divergent mutations impacted cancer relevant genes[21,25] (Fig. 3a,b), similar to previously reported CRC organoids[16]. For example, mutations in EGFR and MLL2 were private to 327_T_PDO, whereas tumour 278_T and its models diverged for mutations in FAM123B, MTOR, and for a PTCH1 frameshift mutation predicted to activate oncogenic sonic hedgehog signalling (Fig. 3b; Supplementary Data 6). Several discordant mutations were either redundant or functionally equivalent. Sample 323 displayed four different mutations in PIK3CA, with R88Q common to both models, A775S private to

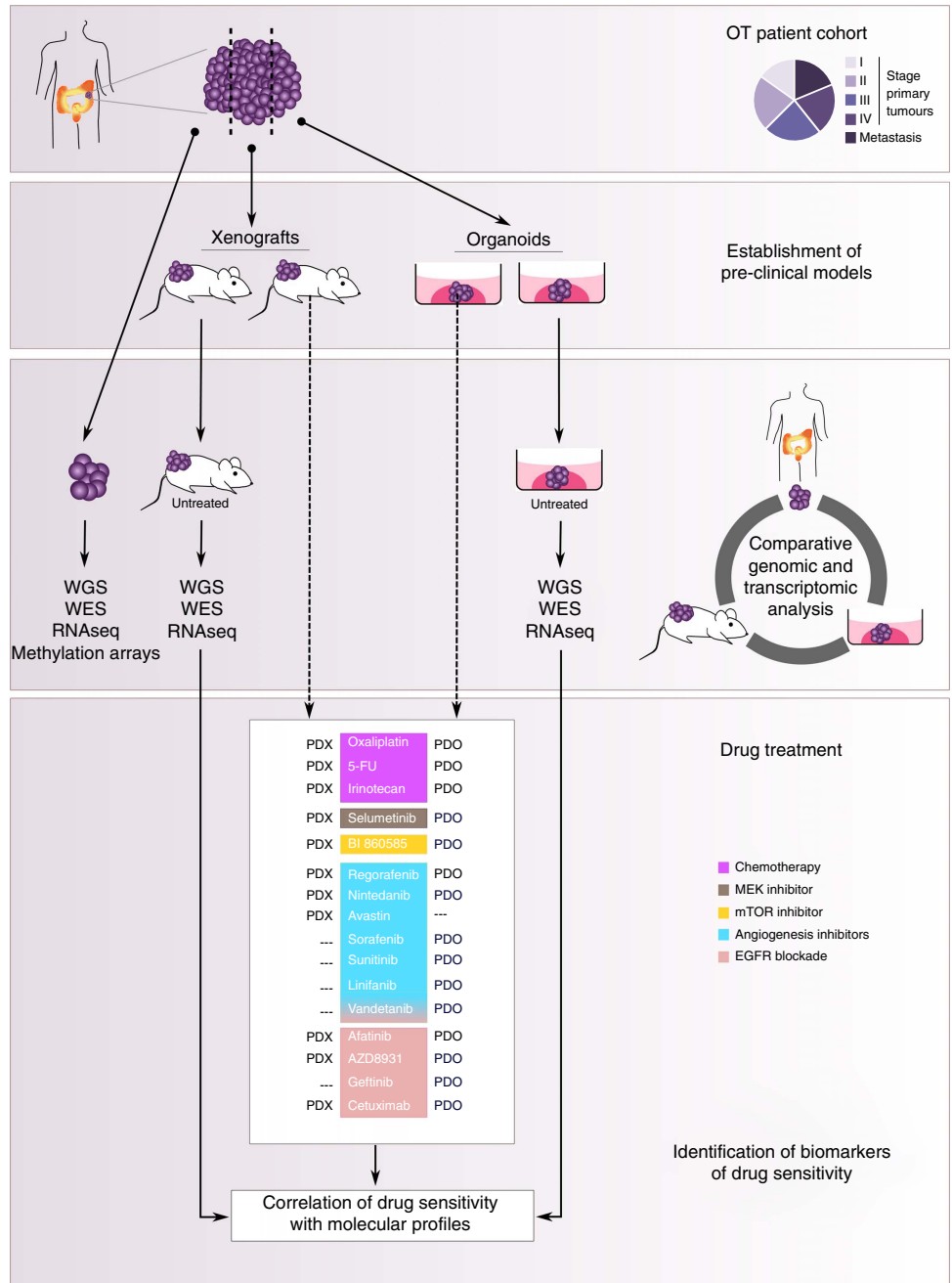

**Figure 1 | Experimental design of the OncoTrack (OT) study.** Resected CRC patient tumours were fragmented and sampled for fuelling the sequencing and the establishment of *in vivo* PDX and *in vitro* PDO models. Untreated original tumours, PDX and PDO samples were analysed by WGS, WES and RNAseq for correlating the molecular information with drug sensitivity patterns. In addition, the epigenomes of the original tumours were analysed. The OT pre-clinical platform treated both model systems with therapeutic compounds representing the standard of care and/or addressing major pathways relevant in CRC.

PDO, whereas G1049R was common to models and donor. Discordant mutations in *APC* were all deleterious, therefore functionally equivalent. These data indicated that pre-clinical models might capture only part of the genetic heterogeneity of the CRC bulk donor tumours, but retained systematically *RAS/RAF* mutations.

**Transcriptome landscapes of tumours and their derived models.** We analysed the global transcriptome profiles of the OT patient tumours and derived models at several levels. Annotation with the

CRC consensus molecular group labels (CMS1 to CMS4) (ref. 2) led to unambiguous classification for only 50/90 patient samples. Further, this classification appeared too coarse for the models, in particular for the organoids (Methods, Supplementary Data 9; Supplementary Fig. 6a,b). Given that we aimed at comparing in details the transcriptome profiles between tumours and model, we analysed *de novo* the RNAseq data from 65 OT primary tumours, excluding at this stage low purity (<40%) or metastatic samples to minimize confounding contribution of surrounding tissues (see methods). In a first step, we applied non-negative matrix factorization (NMF)[26] and CLICK[27] algorithms. NMF identified

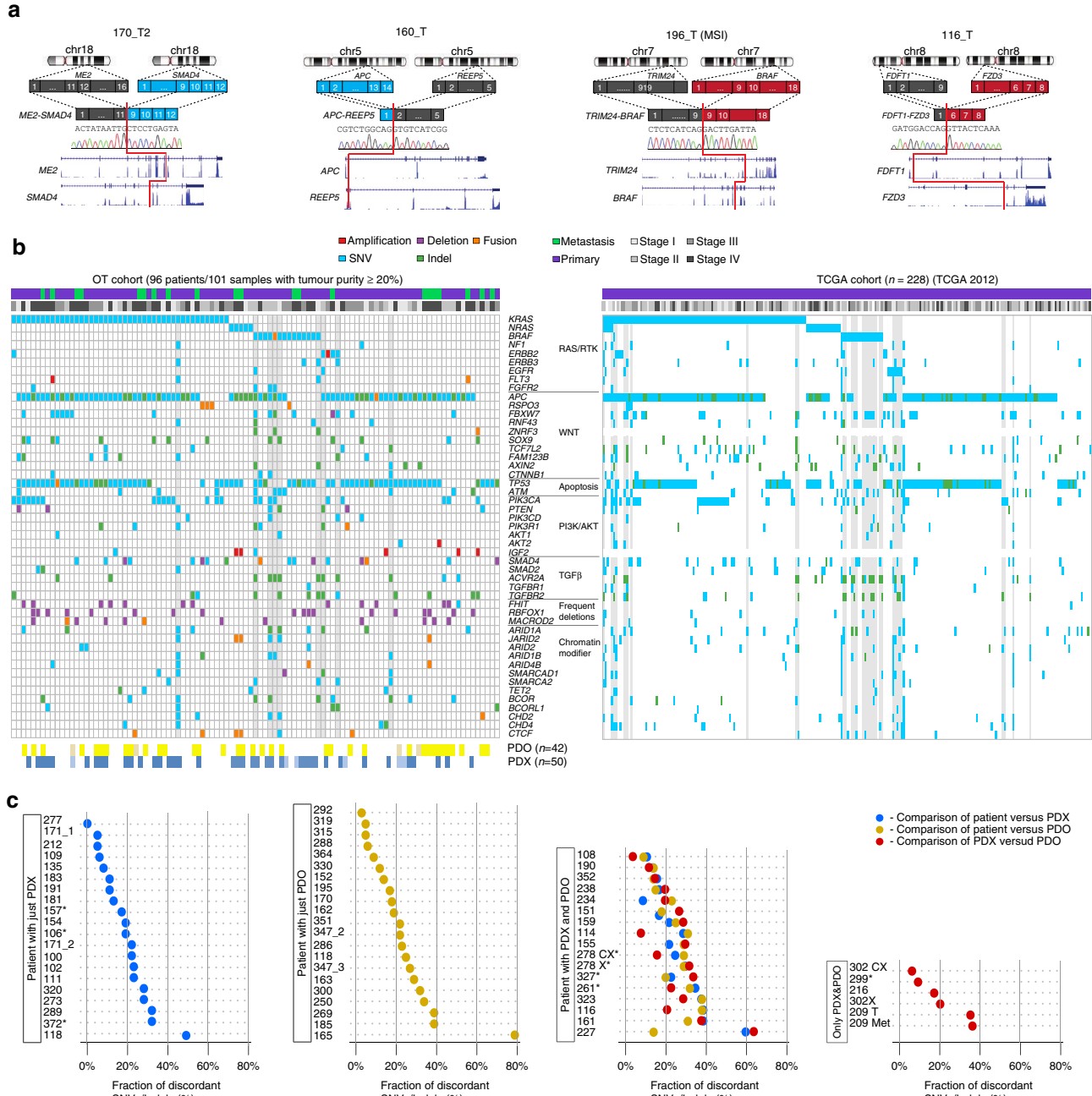

**Figure 2 | Genomic landscape of the OT patient and model cohorts. (a)** Examples of gene fusions, either deleterious (*ME2-SMAD4* and *APC-REEP5*, blue) or activating (*TRIM24-BRAF* and *FDFT1-FZD3*, red); schematics display chromosomal location, fusion partners, exon structure (blue, red and grey filled boxes), fusion breakpoint (red line), validation by Sanger sequencing and RNA read coverage. **(b)** Landscape of the most recurrent somatic alterations in 101 OT primary and metastasis samples (tumour purity of ≥20%) from 96 patients compared with the mutation pattern in 228 primary CRC tumours from TCGA (ref. 25) (SNV + Indel). Genes are grouped according to biological pathways. MSI and hypermutated samples are shaded in grey. Tumour stages are coded in grey shades on top of the matrix. Metastasis and primary samples are indicated respectively in purple and green boxes on top of the matrix. Alteration types are colour-coded as indicated. Obtained PDX and PDO models are depicted in blue and yellow, respectively at the bottom of the matrix. Darker shades correspond to the models shown in Fig. 2c. **(c)** Fraction of damaging and expressed somatic SNVs/Indels found discordant between original patient tumours and matched models, comparing 56 patient samples (tumour purity ≥40%) and their corresponding *ex vivo* models (37 PDX and 37 PDO), as well as five PDX-PDO siblings without matching patient samples. Dot colours indicate the following: patient versus PDX (blue), patient versus PDO (yellow) or PDX versus PDO (red). Samples described in Fig. 3b are marked in bold. MSI and hypermutated samples are marked with an asterisk. Panels from left to right: patients with an established PDX only; patients with an established PDO only; patients with established PDX and PDO; PDX/PDO siblings without a corresponding sequenced patient tumour. In the cases of tumours 278 and 302, the suffix 'X' represents patient-derived PDX and 'CX' represents PDO-derived PDX.

three main groups, OT_NMF1, OT_NMF2 and OT_NMF3 (Supplementary Data 10; Supplementary Fig. 7a), defined by tumours sharing global biological features (Supplementary Data 11). CLICK identified 13 gene signatures (OT_C1 to OT_C13) (Supplementary Data 10; Supplementary Fig. 7b) corresponding to co-expressed genes in specific tumour sub-

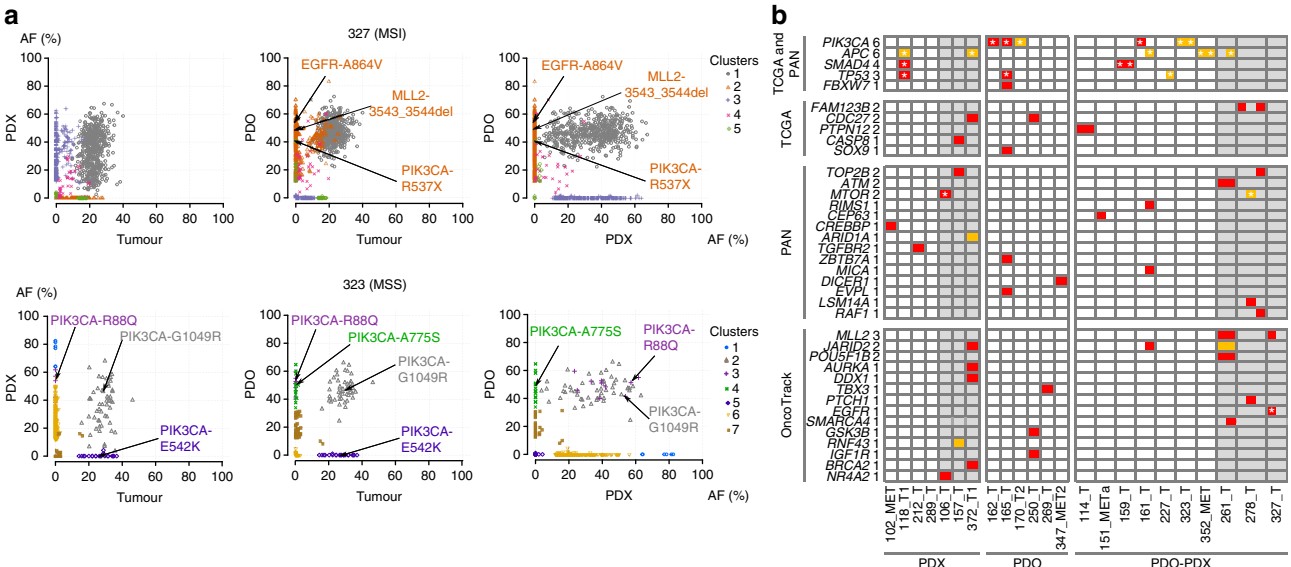

**Figure 3 | Molecular discordances between original tumours and derived models in CRC-relevant genes.** (**a**) Examples of clonality analysis based on the sciClone algorithm, for samples 327_T (MSI—40% tumour purity) (upper panel) and 323_T (MSS—60% tumour purity) (lower panel). The plots display the mutations in diploid regions clustered by their allele frequencies in: original tumour versus PDX (left), original tumour versus PDO (middle) and PDX versus PDO (right). Individual mutation clusters were shown by different colours where grey indicated the mutations found in common between the compared samples. The PDO derived from 327_T displayed private *EGFR*, *MLL2* and *PIK3CA* mutations in cluster 2. The PDX derived from 323_T displayed different clones harbouring private mutations in *PIK3CA*. (**b**) Somatic clonal mutations discordant between patients/PDX (left), patients/PDO (middle) and patients/PDX/PDO (right). Genes found mutated only in either tumour or model (or in only one model) are marked by red squares. Genes with different mutations in tumours and models are marked by orange squares. Mutations validated by targeted sequencing are indicated with asterisk. MSI and hypermutated samples are shaded in grey: Only cancer-relevant genes are shown, selected from TCGA, PAN or OT recurrently mutated genes resources (see methods). Patients with a purity of ≥40% are shown.

groups, leveraging the depth of functional annotation attributed to tumour groups (Supplementary Data 11). We rationalized the NMF and CLICK clusters in a meta-analysis of 38 gene signatures incorporating previous CRC molecular groups[1,3–5]. The mean expression values for these 38 signatures were calculated for 90 OT samples, now including metastases (tumour purity ≥40%). Analysis of this matrix with unsupervised hierarchical clustering and mclust[28] identified three main molecular groups with high stability referred to herein as ASCL2/MYC, ECM/EMT and Entero/Goblets (Supplementary Fig. 8a–c), respectively related to CMS2, CMS4 and CMS3 (ref. 2), albeit with differences (Supplementary Fig. 8d). MSI samples (CMS1) clustered within Entero/Goblets. ASCL2/MYC (OT_NMF2) featured higher expression of *ASCL2*, the master regulator of colonic crypt stem cells[29], *MYC* and *AURKA* potentially contributing to stem cell phenotypes[30], and *CDX2* promoting Wnt signalling (Supplementary Fig. 9). This group showed a trend for wild-type *BRAF* and *MACROD2* deletions (Fisher's exact test, $P = 0.01$ and 0.02, respectively) and was associated with a specific DNA methylation pattern, namely cluster_4 ($P = 0.0057$, Chi-square test) (Supplementary Fig. 10). ECM/EMT (OT_NMF1) was characterized by TGFβ and sonic hedgehog signalling, extracellular matrix (ECM), epithelial to mesenchymal transition (EMT), inflammation, enteric neurons and smooth muscle cell markers reflecting tumour invasion in the intestinal myoenteric layer[31,32]. Entero/Goblets (OT_NMF3), associated with the CIMP-High methylation pattern (Supplementary Fig. 10), had strong features of epithelial enterocytes, enteroendocrine and goblets cells, carbonate dehydratase activity, and higher levels of *AGR2*, known to promote adenocarcinomas growth[33] (Supplementary Fig. 9). Entero/Goblets were mainly early stage tumours whereas ASCL2/MYC and ECM/EMT were rather

late stages (Supplementary Fig. 8e). Data highlighted different stromal and immune environments among tumours of the same group (Supplementary Fig. 8b–8f). Entero/Goblets tumours were associated with Th17 cytokines, where MSI cases were unique in expressing innate immunity signatures (for example, OT_C8) (Supplementary Fig. 11a,b). ASCL2/MYC tumours displayed the lowest immune infiltration, in contrast to the highly inflamed tumours of the ECM/EMT group showing a maximal tumour purity of 50% (Methods, Supplementary Fig. 11a,b and Supplementary Data 12).

By comparison, models defined only two main molecular groups, either NMFa strongly enriched for Wnt and stemness processes pointing to ASCL2/MYC, or NMFb corresponding to colonic epithelial cells and entero/goblets (Fig. 4a, Supplementary Fig. 12a). Both *in vitro* and *in vivo* model systems lacked ECM, stromal components and human immune-related signatures (Fig. 4a, Supplementary Data 12), whereas showing prominent aurora kinase pathway/crypt progenitors, lipid metabolism and oxidative respiration signatures (OT_C2, OT_C4 and OT_C10), in part because of their higher tumour content. However, the two model types exhibited significant differences (Fig. 4b). Within the limits of the missing human stroma, the main features of the donor tissues were preserved in PDXs where ASCL2/MYC- and ECM/EMT-derived xenografts grouped in NMFa, whereas those engrafted from Entero/Goblets were NMFb. PDOs showed a more complex picture (Fig. 4c), where ASCL2/MYC-derived cells mostly featured the NMFb pattern. Despite the limited number of sibling models with this phenotype, data suggested heterogeneity/plasticity in cell cultures. As a proof of principle, we combined a Wnt reporter assay with RNAseq on PDOs derived from 151_MET1 (ECM/EMT). Data demonstrated the co-existence of two genetically identical cell sub-populations, respectively

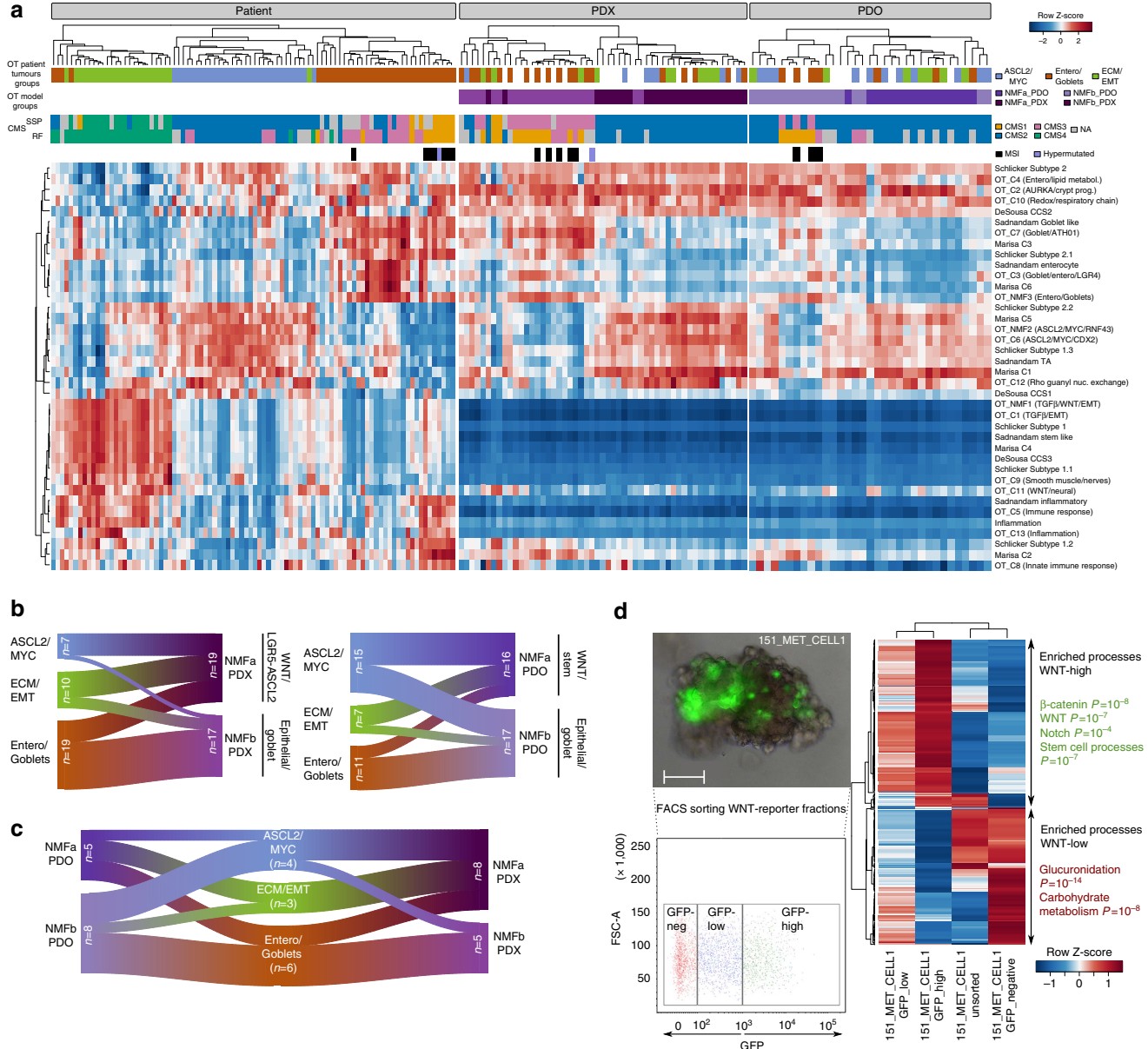

**Figure 4 | Expression profiling of the OT patient (tumour purity ≥40%) and model cohorts.** (**a**) Unsupervised hierarchical clustering of the mean patterns of 38 CRC-related signatures in 90 patients, 53 PDX and 33 PDO models. Information on molecular groups is indicated as follows from top to bottom under the dendrogram. Colour labels of the three OT patient tumour groups, NMF classes for the models, and CMS groups (RF: upper row, SPP: bottom row) are indicated in the caption on the right side. MSI- and hypermutated samples are marked with black and blue boxes, respectively. (**b**) Sankey plots showing the correspondence between the molecular groups in the patient tumours (ECM/EMT, ASCL2/MYC, Entero/Goblets) and the two NMF groups in PDX and PDO models. (**c**) Sankey plots showing the correspondence between the NMF molecular groups between sibling pairs of PDX and PDO models. (**d**) Analysis of WNT-signalling reporter assays on 151_MET PDO cultures. Left: cell cultures showing the GFP green fluorescent signal revealing WNT signalling and FACS-sorting of the corresponding GFP-high, -low and -negative cell fractions. Scale bar is 100 μm. Right: Heatmap representing differentially expressed genes (n = 404) between the GFP-low, GFP-high, GFP-negative cell fractions and unsorted cells. Genes were filtered by fold change (FC) and difference of RPKM (DR, |DR|≥1, |log₂(FC)|≥2). Key enriched processes and P-values in WNT-negative and -high fractions are indicated on the right side.

presenting either high-Wnt signalling or low Wnt with epithelial features (Fig. 4d; Supplementary Data 13).

Sibling pairs of PDX/PDO differentially expressed signatures for hypoxia, EMT, G2M checkpoints and proliferation (Fig. 5a), as well as stemness markers (Supplementary Fig. 13a), without a trend for a given model type. However, PDOs had significantly higher levels of the stem cell marker *ALDH1A1* (Supplementary Fig. 13b) and of components of carbohydrate, steroid, retinoid

and fatty acid metabolism (Fig. 5b). *SCD* encoding Stearoyl-CoA desaturase-1, a key enzyme in fatty acid metabolism involved in cancer cell survival[34], and UGTs (UDP-glycosyltransferases) triggering glucuronidation activating lipid metabolism and mediating drug resistance[35] were more active in organoids than in 2D CRC cell lines (Fig. 5c). This might be because of metabolic adaptation to the culture conditions or to intrinsic features of organoids.

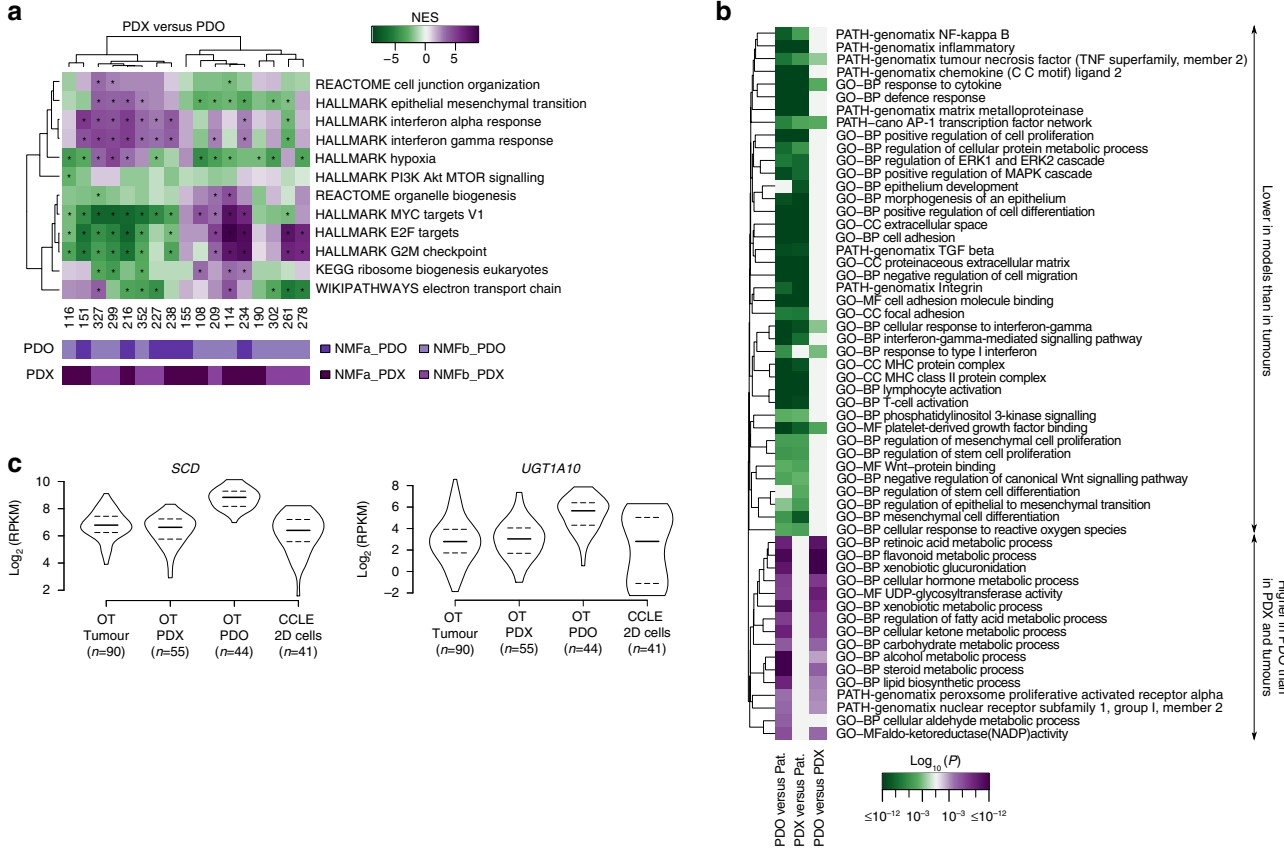

**Figure 5 | Comparative gene expression profiling between PDO and PDX models.** (**a**) Differentially expressed canonical pathway gene signatures. Positive or negative normalized enrichment scores (NES) calculated for PDX versus PDO for each given pathway are indicated in purple or green shades, respectively. Asterisks indicate significant enrichment with FDR < 0.005. The gene set enrichment was determined using a pre-ranked GSEA. Genes were ranked according to their expression fold change and expression difference between the corresponding model pairs. (**b**) Differential expression of gene signatures for canonical pathways and GO terms in PDO versus tumour, PDX versus tumour and PDO versus PDX. Green and purple boxes indicate $\log_{10}(P\text{-values})$ for down- or up-regulated gene sets, respectively. Tumour samples with a tumour purity $\geq 40\%$ are shown. (**c**) Violin plots showing the expression of *SCD* and *UGT1A10* in the OT cohorts (patient tumour purity $\geq 40\%$, PDX and PDO) and CCLE 2D colorectal cancer cell lines (see methods). Lines mark the 25%, 50% and 75% quantiles.

**Patterns of drug response in CRC-derived models.** We monitored the sensitivity of 94 models to 16 drugs, chemotherapeutics and drugs targeting MEK, mTOR, VEGFR, or EGFR pathways (Figs 1 and 6). In PDOs, we considered efficacy ($E_{\text{max}}$), the maximum growth inhibition reached at the highest drug concentration, as well as potency, the half-maximum inhibitory concentration ($IC_{50}$) relative to the reference compound staurosporine, which exhibited low $IC_{50}$/high $E_{\text{max}}$ (Fig. 6a,b; Supplementary Data 14). Potency and efficacy were highly correlated for all drugs in all organoids (Supplementary Fig. 14a,b). Potency was used to define four response categories, strong-, moderate-, minor response or resistant (Supplementary Fig. 14c), as shown for AZD8931 (Supplementary Fig. 14d). PDOs exhibited a wide range of sensitivities upon treatment with the different drugs (Supplementary Figs 14 and 15a). Chemotherapeutics achieved only poor responses, particularly oxaliplatin which was ineffective. Irinotecan was used here instead of its active derivative SN-38, given that the cells expressed *CES2*, essential for metabolizing the pro-drug. Responses to irinotecan were marginal, as previously observed in organoids[16], potentially because of glucuronidation inactivating SN-38 in organoids. Multi-kinase inhibitors (MKIs) triggered minor responses, (Fig. 6a,b). The sensitivity profiles to linifanib, nintedanib and sunitinib were strongly correlated, but less so to sorafenib and regorafenib, a

recently approved MKI for metastatic CRC (ref. 36), structurally similar to each other and found together in a different cluster (Fig. 6c). Vandetanib, targeting both VEGFR and EGFR, was the most potent MKI but correlated strongly with EGFR blockade, indicating that the anti-proliferative effects were likely mediated by the EGFR pathway inhibition. Responders to EGFR blockade were mostly *KRAS/BRAF* wild-type or carried BRAF-G466V showing reduced kinase activity[37] (Supplementary Fig. 16a). EGFR blockade displayed the widest range of effects across samples, as previously observed[38] (Fig. 6a,b; Supplementary Fig. 15a). Cetuximab was ineffective in cells at clinically relevant concentrations (Supplementary Fig. 15b; Supplementary Data 14), corroborating earlier reports[39] and data streaming from an independent CRC organoid cohort estimating IC50 in the mmolar range[16], although biologically meaningful concentrations of cetuximab remain in the nmolar window[40]. Further, only few *KRAS/BRAF* wild-type CRC 2D cell lines respond to cetuximab[14]. In contrast, PDOs were sensitive to tyrosine kinase inhibitors (TKIs), in particular to afatinib, and their response to AZD8931 was similar to earlier studies[16]. The model 330_T_CELL with *ERBB2* amplification was sensitive to AZD8931 (targeting EGFR, ERBB2, ERBB3) and afatinib (targeting EGFR, ERBB2), but less so to gefitinib targeting only EGFR (Supplementary Fig. 16a). PDO 327_T_CELL had a rare

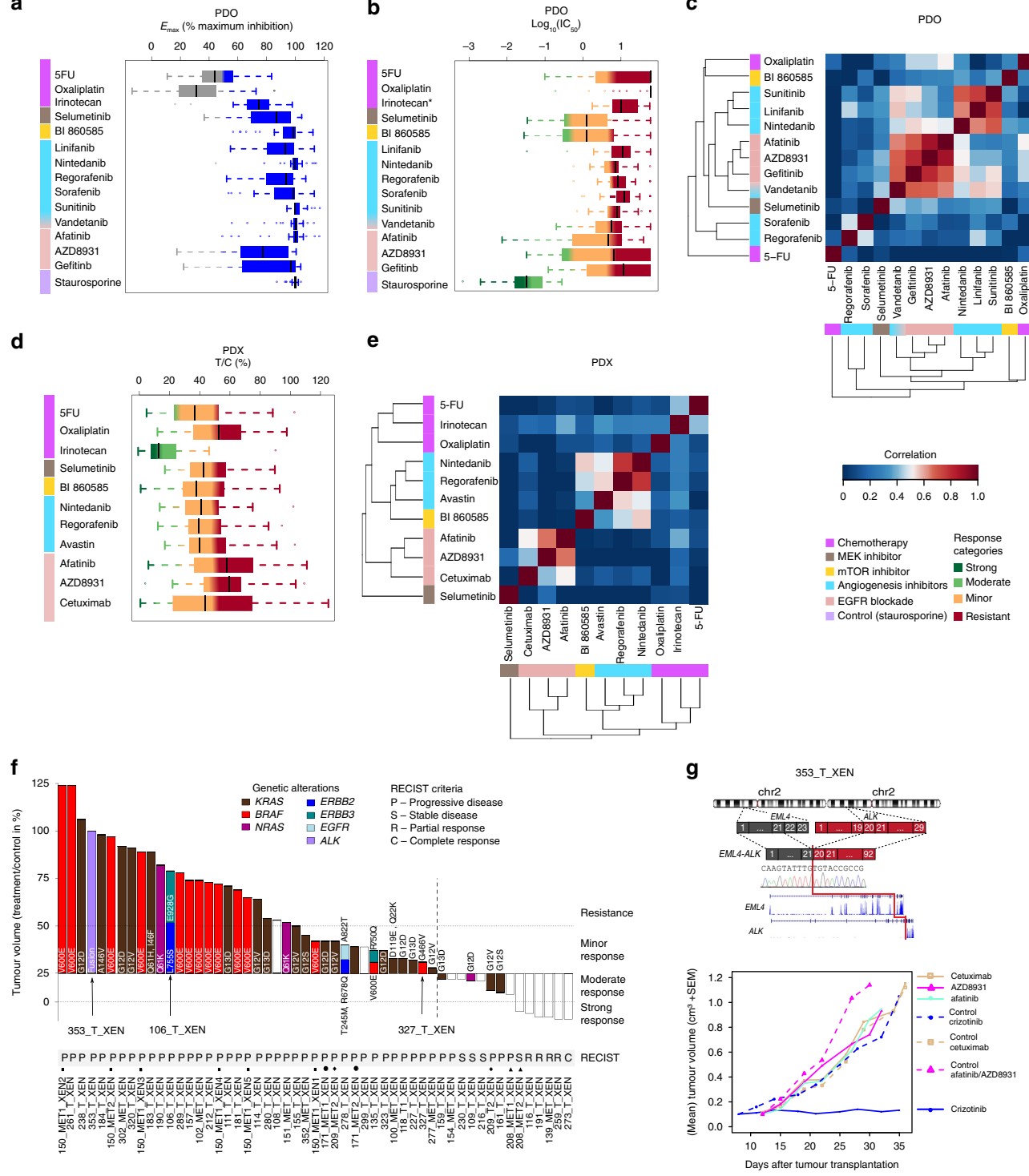

**Figure 6 | Overview of drug response in PDX and PDO models.** (**a**) Box plot showing the maximum inhibition ($E_{max}$) values for 14 drugs and for the staurosporine control in 35 PDO models. Irinotecan was tested on 21 PDOs. Range of drug efficiency ($E_{max} > 50\%$) or inactivity ($E_{max} < 50\%$) is indicated in blue and grey, respectively. (**b**) Box plot showing the potency of 14 drugs and of the staurosporine control based on $\log_{10}(IC_{50})$ in 35 PDO models. The four response categories are coloured as indicated in the caption. (**c**) Hierarchical clustering representing the correlation of drug sensitivity patterns in PDOs based on the $\log_{10}(IC_{50})$ values. (**d**) Box plot showing the sensitivity of 57 PDX models to 11 drugs based on tumour/control volumes (T/C in %) values: response categories are coloured as indicated in the caption. (**e**) Hierarchical clustering representing the correlation of drug sensitivity patterns based on T/C values in PDXs. (**f**) Waterfall plot showing cetuximab sensitivity for 53 PDX and indicating mutations and fusions in RAS/RAF and RTK pathways. Both T/C and RECIST results are indicated; models derived from the same patient are labelled with the same symbol above each sample name. The vertical dash line shows the cut-off between responders and non-responders. (**g**) Analysis of the activating *ALK* gene fusion in 353_T_XEN; upper part: schematics displaying chromosomal location, fusion partners, exon structure (grey and red boxes), fusion breakpoint (red line), validation by Sanger sequencing and RNA read coverage. Lower part: treatment response of OT PDX 353_T_XEN to crizotinib, cetuximab, AZD8931 and afatinib: mean tumour volume of treated and corresponding controls measured between day 8 and day 36 after treatment.

*EGFR* mutation (A864V) and was highly sensitive to both AZD8931 and afatinib. Treatment with MEK or mTOR inhibitors achieved objective responses but PDOs displayed different drug response patterns (Fig. 6a–c).

For xenografts, drug response was assessed by the relative tumour volume of the treated PDX versus its matched untreated control (T/C), and classified into four categories strong-, moderate-, minor response or resistant (Fig. 6d, Supplementary Data 14). Among chemotherapeutic agents, the outstanding response to irinotecan was uninformative given the intense metabolic activation of the pro-drug in the mouse[41]. In contrast, the anti-metabolite drug 5-FU, that has comparable serum pharmacokinetics in human patients and nude mice[42], was the most efficient chemotherapy in PDXs (Fig. 6d; Supplementary Fig. 15c). The three compounds targeting angiogenesis triggered objective responses (Fig. 6d). Mechanistically, *VEGFA* was expressed in PDXs, but not the human *VEGF* receptors (*FLT1*, *FLT3*, *KDR*), which were however replaced by the murine compartment (Supplementary Fig. 17), supporting active VEGF signalling via murine endothelial cells. The response pattern to regorafenib[43] correlated strongly with nintedanib targeting VEGFR, but only weakly with Avastin, a monoclonal antibody inhibiting human VEGFA (Fig. 6e). Interestingly, targeted inhibition of mTOR (mTORC1/C2) correlated with nintedanib and regorafenib response profiles (Fig. 6e), which might be mechanistically related to the contribution of mTORC2 as a critical signalling node for VEGF-mediated angiogenesis[44]. The MEK inhibitor selumetinib displayed a response profile different to that of all other drugs.

The EGFR blockade achieved tumour growth inhibition >50% for one third of the xenografts (Fig. 6d, Supplementary Fig. 15c, Supplementary Data 14). Responses to TKIs targeting the intracellular domain of EGFR and other ERBB family members were comparable with each other but less so with the selective EGFR antibody cetuximab, performing best in achieving a substantial growth delay in 10% of the xenografts (Fig. 6d–f). *BRAF* V600E was the most stringent independent predictor of resistance, as previously reported[45]. *RAS* mutations appeared less specific based on the T/C criteria, with four PDXs carrying *KRAS* or *NRAS* mutations among the moderate responders. However, when applying the RECIST clinical criteria, some of the responders were considered progressive (Fig. 6f, Supplementary Fig 16b). This apparent discrepancy reflects the fact that RECIST records whether progression exceeds 20% of the initial tumour volume after treatment, whereas T/C compares the relative tumour volume of the treated tumour versus its untreated matched control, thus permitting to detect objective tumour growth delay, even minor (Fig. 6f). Similar cases of non-progressive *KRAS* mutated tumours were already reported for patients treated with cetuximab[46,47]. In some cases, we could mechanistically link the response profile to EGFR blockade to specific mutations. 106_T_XEN carrying *ERBB3*-E928G and the activating mutation *ERBB2*-L755S conferring sensitivity to ERBB2-targeted drugs but resistance to EGFR inhibitors[48] showed indeed objective response to afatinib and AZD8931 but resistance to cetuximab (Fig. 6f; Supplementary Fig. 16b). PDX 353_T_XEN carrying an *EML4-ALK* fusion as sole putative driver was resistant to all EGFR inhibitors (Fig. 6f, Supplementary Fig. 16b), but showed exquisite sensitivity to crizotinib (Fig. 6g), approved for targeting *ALK* fusions in lung cancer[49]. A similar fusion responding to crizotinib was described for one CRC cell line[14]. *ALK* fusions are found in ~1% of the CRCs[50]. We present the first *in vivo* pre-clinical data suggesting that crizotinib might represent a therapeutic option for CRC patients with rare *ALK* fusions.

**Comparative drug responses between PDX/PDO sibling pairs**. We compared the treatment outcomes of eight drugs for 19 pairs of PDO/PDX siblings (Fig. 7; Supplementary Fig. 15d), although

inherent differences in the microenvironment (matrigel versus vascularized tissues), biology of the models, oxygen levels, catabolism/anabolism of the compounds *in vitro* and *in vivo*, and criteria assessing drug sensitivity ($IC_{50}/E_{max}$ versus T/C) are challenging this exercise. Irinotecan and cetuximab were excluded since they were uninformative in PDXs or in PDOs. Relying on the four response categories defined above, we called concordant response between PDX/ PDO siblings if their responses did not differ by more than one rank (for example, minor versus moderate) (Fig. 7, blue and white areas). Despite the aforementioned contingencies, response patterns between the two systems were fairly concordant excepted for AZD8931 and 5-FU. For the EGFR blockade, afatinib responses were in reasonable agreement but less so for AZD8931 more potent in PDOs than in PDXs, despite its low efficiency in cells (Fig. 7). This is likely to be because of differences in the pharmacokinetic behaviour of this compound, to biological differences between models, or to genetic heterogeneity. Models derived from 327_T differed for EGFR-A864V private to PDOs and absent in PDXs, which showed high-sensitivity or resistance to EGFR inhibitors, respectively (Supplementary Fig. 15d). 5-FU was the second most discrepant drug (Fig. 7) possibly because of its complex metabolism. 5-FU is catabolized *in vivo* via the dihydrothymine dehydrogenase (DPYD/DPD), so that only 1–3% of the initial 5-FU concentration leads to active metabolites in plasma, generating modified nucleotides that are incorporated into the RNA and DNA via different enzymatic reactions[51]. Despite the fact that PDOs do not express DPD, the $E_{max}$ was <50%, suggesting that anabolic routes might be less efficient *in vitro*.

**Molecular classifiers of drug response**. Exploiting the high dynamic range of RNAseq, we performed differential gene expression analyses for identifying molecular profiles associated with drug response profiles (see Methods). This approach was successful for 5-FU, Avastin and EGFR inhibitors (Supplementary Data 15), but failed for the other drugs in part because of their low efficacy precluding the assessment of true responder groups. However, we were also unable to identify common features characterizing responders to the efficient compound BI 860585, inhibiting both mTORC1 and mTORC2, suggesting that this drug might trigger complex cascade effects, difficult to capture in our present analysis.

For 5-FU, we focused on PDXs more comparable to the patient situation[42]. Responding xenografts tended to belong to NMFb (Fisher's exact test $P = 0.03$) and included the MSIs (Fig. 8a). We identified a discriminative gene signature characterized by intestine epithelial lineage ($P = 1 \times 10^{-8}$) and HIF-1 transcription network ($P = 1 \times 10^{-7}$). Combined with recursive feature elimination and support vector machine (SVM) algorithms, this signature led to build a mini-classifier of 14 genes able to segregate responders from non-responders (Methods, and Fig. 8b). The stability of the classifier was estimated by cross-validation on the PDX OT-cohort (Supplementary Data 16). On the OT-patient cohort, the 5-FU classifier predicted that the best responders would be from the Entero/Goblets molecular group, including 7/8 MSI-high samples (Fig. 8c; Supplementary Data 16). However, the predictive power of the classifier will require further assessments on additional cohorts treated exclusively with 5-FU. Considering inhibitors of angiogenesis, only the response to the antibody Avastin blocking VEGFA could be associated with a clear molecular signature, although the limited number of responders impeded statistical assessments. Sensitive tumours exhibited higher activity of genes involved in energy-coupled mitochondrial transport ($P = 1 \times 10^{-4}$), whereas resistant tumours exhibited strong colonic cell features and higher expression of *ERF* (Supplementary Fig. 18a).

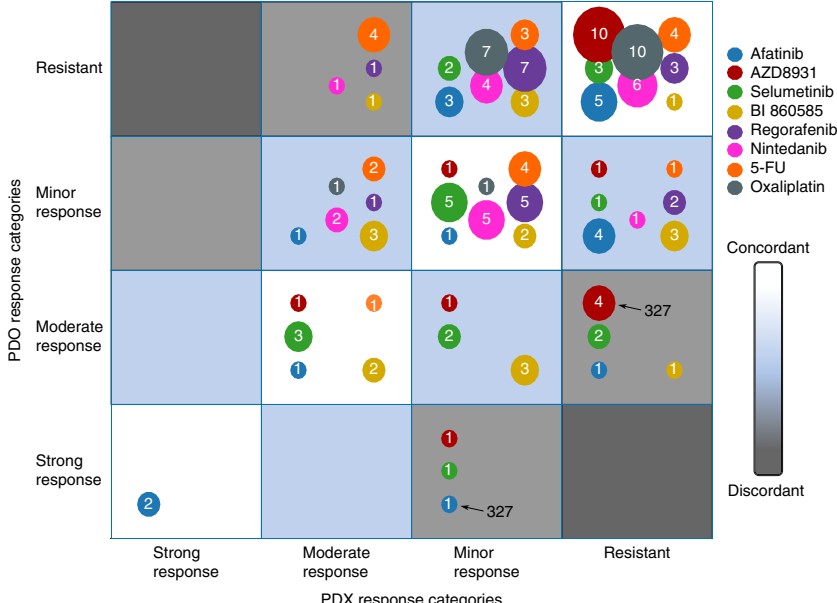

**Figure 7 | Bubble plot comparison of drug sensitivity profiles in 19 sibling pairs of PDX and PDO models.** Drug response categories are displayed on the axes (x = PDX, y = PDO). The size of each bubble is proportional to the number of model pairs in a given response category, where this number is indicated in the bubbles. The colours represent the eight drugs tested in both model systems, as indicated in the caption. The background colour in each square indicates the degree of concordance between the two models where grey indicate discordant drug response.

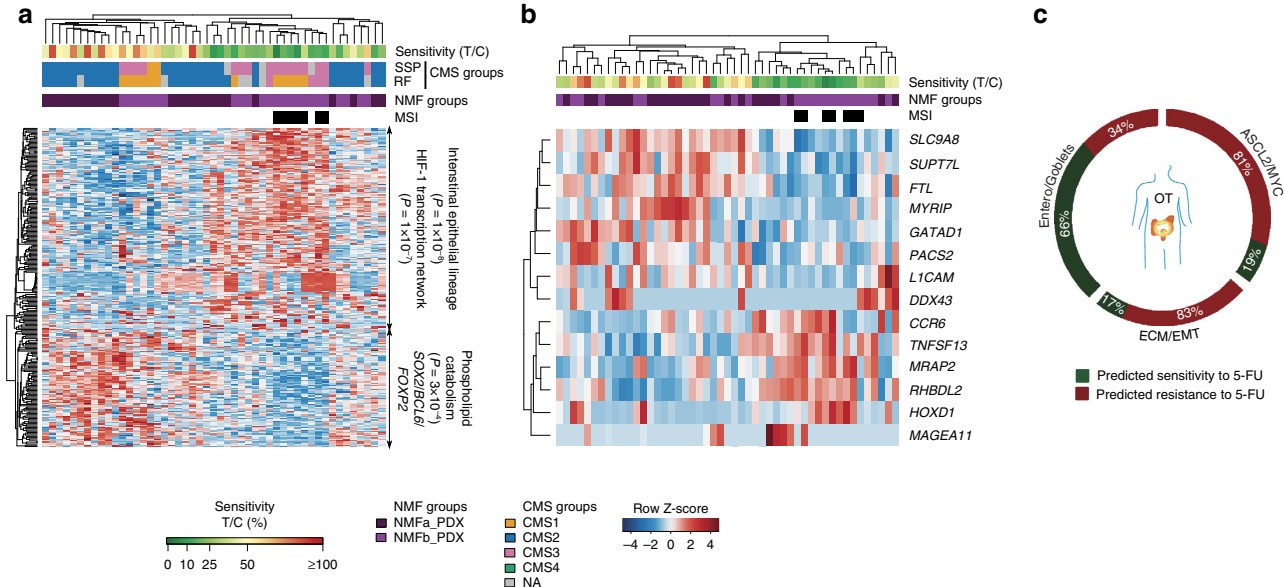

**Figure 8 | Molecular classification of response to 5-FU in PDX models.** (**a**) Hierarchical clustering of PDX samples based on the 5-FU response gene signature ($n = 251$ genes). The drug sensitivity is given by T/C continuous values coloured as indicated in the caption (dark green: strong response, red: resistance). MSI samples, OT-NMF-groups and CMS classifications are coloured as indicated in the caption. (**b**) Hierarchical clustering of PDX samples based on the mini-classifier ($n = 14$ genes) obtained by SVM-based machine learning on 49 PDX. (see Supplementary Data 14 for cross-validation). (**c**) Distribution of the predicted responders and resistant patient tumours (tumour purity $\geq 40\%$) in the three molecular groups (ECM/EMT, ASCL2/MYC, Entero/Goblets).

Response patterns to EGFR inhibitors were associated with characteristic molecular profiles, albeit different ones in PDOs and in PDXs (Fig. 9, Supplementary Fig. 18b, Supplementary Data 15). PDOs sensitive to afatinib featured enterocyte differentiation factors ($P = 1 \times 10^{-6}$) such as *ATOH1*, and higher expression of *EGF*, and *GREM1*, which promotes stem cell properties in colonic cells outside of the stem cell niche[52]. Response to AZD8931 defined a similar profile to afatinib, in contrast to gefitinib and vandetanib, which were less efficient (Figs 6b and 9a,b; Supplementary Fig. 18b). Fourteen genes were associated with resistance to all EGFR inhibitors (Supplementary Fig. 18c–e), including *RGS4*, *BASP1* and *IGF2* known as marker of EGRF blockade insensitivity[53].

In PDXs, strong responders to the EGFR blockade were mostly from molecular group NMFa (stem/Wnt) whereas resistant ones

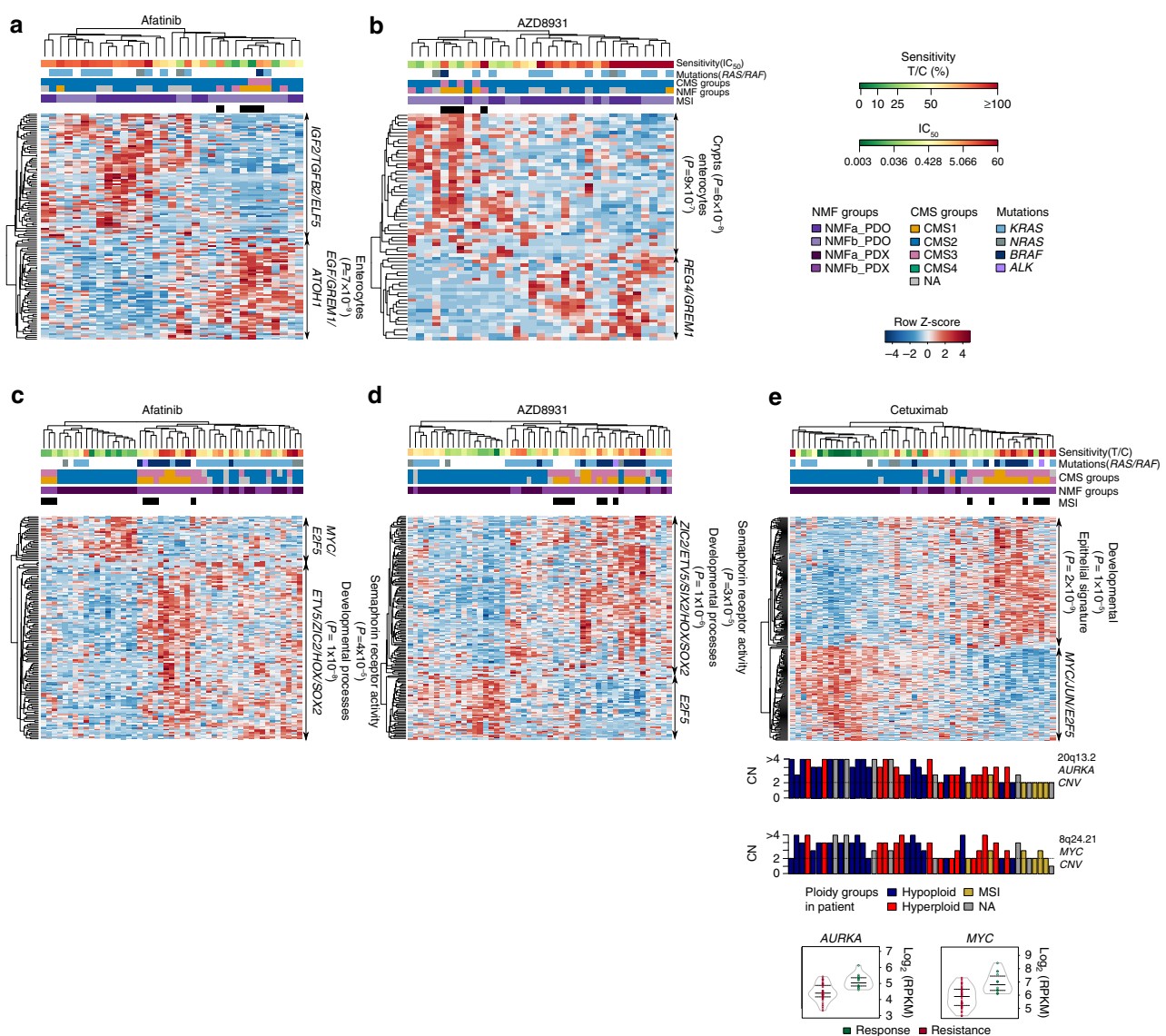

**Figure 9 | Molecular classification of response to EGFR inhibitors in PDX and PDO models.** Drug sensitivity is represented by T/C (PDX) or IC$_{50}$ (PDO) continuous values coloured as indicated in the caption (dark green: strong response, red: resistance). MSI samples, OT-NMF-groups and CMS classifications (SSP: upper row, RF: bottom row) and mutation status are coloured as indicated in the caption. (**a,b**) Hierarchical clustering of 33 PDO samples based on the drug response gene signatures for afatinib (**a**) or AZD8931 (**b**). (**c,d,e**) Hierarchical clustering of 49 PDX samples based on the drug response gene signatures for afatinib (**c**), AZD8931 (**d**) or cetuximab (**e**). Copy numbers (CN) of *MYC* (8q24.21) and *AURKA* (20q13.2) loci in PDX samples are shown below the cetuximab signature. The bars are colour-coded according to the CNV groups in the patients (Hypoploid, Hyperploid and MSI). Violin plots show the *AURKA* and *MYC* expression in responders versus resistant PDXs (response T/C ≤ 25%, resistance T/C > 25%). The expression values are shown as log$_2$ RPKM. Lines indicate the 25%, 50% and 75% quantiles, respectively.

were epithelial (Fig. 9c–e). Resistance to AZD8931 and to afatinib was associated to a molecular signature enriched for developmental genes ($P = 1.03 \times 10^{-6}$ and $1 \times 10^{-9}$), including the oncogene *ETV5* (Supplementary Data 15) and for AZD8931, *REG4*, a multifunctional mitogenic protein known as a potent activator of the EGFR pathway in CRC (refs 54,55). Response profiles to cetuximab showed distinctive signatures segregating strong responders from resistant tumours, showing also a third group of PDXs displaying moderate/minor responders that included *KRAS* mutants (Fig. 9e). Resistance to cetuximab was also associated with expression of *REG4*, and featured strong epithelial ($P = 2 \times 10^{-9}$), as well as developmental ($P = 1 \times 10^{-5}$) signatures whereas sensitive tumours featured expression of *MYC*, *JUN* and *E2F5*, and of previously reported markers of cetuximab sensitivity *CEACAM7* and *EREG*[46,56].

Sensitivity was correlated with hypoploidy (Fisher's exact test $P = 0.003$), and with CNV gain of *MYC* and *AURKA* (Fig. 9e), suggesting that tumours sensitive to EGFR blockade might rely on the MYC-AURKA axis, a relevant pathway in CRC (ref. 57).

The EGFR response signatures of PDX and PDOs shared only a few genes, highlighting their biological differences and the difficulties in comparing directly these systems (Supplementary Fig. 18f,g).

**Classifier of cetuximab sensitivity in *KRAS*-WT tumours.** Given that a fraction of patients with *BRAF/KRAS/NRAS* wild-type (WT) CRCs do not benefit from cetuximab, we aimed at identifying an independent classifier predicting cetuximab sensitivity for addressing this unmet clinical need. On the basis of the

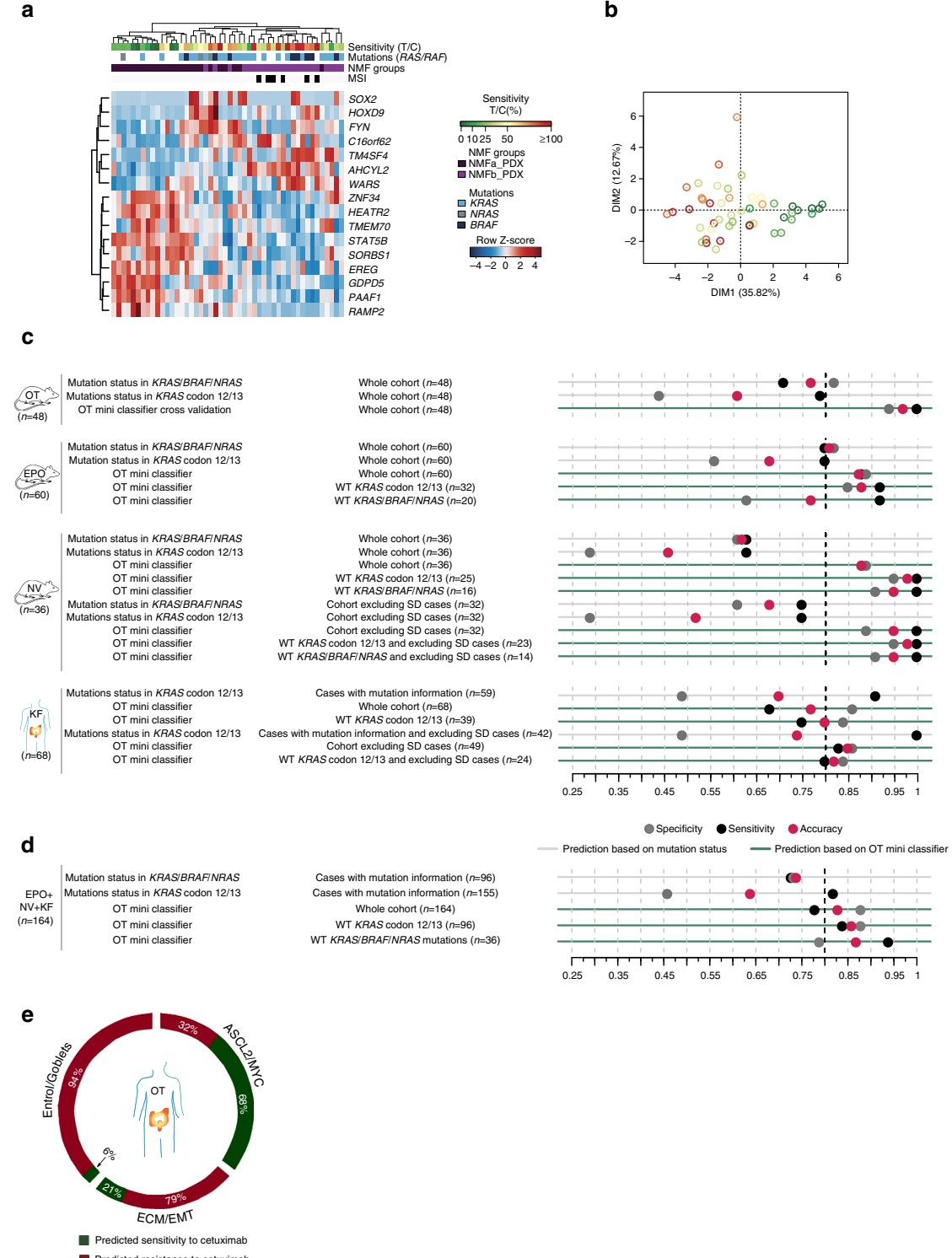

**Figure 10 | Validation of the mini-classifier of cetuximab response in PDX.** The drug sensitivity is given by T/C continuous values coloured as indicated in the caption (dark green: strong response, red: resistance). MSI samples, OT-NMF-groups, CMS classification and mutations are coloured as indicated in the caption. (**a**) Hierarchical clustering of 48 PDX samples based on the mini-classifier (n = 16 genes) obtained by SVM-based machine learning. (**b**) Principal component analysis (PCA) showing the classifier's ability to separate responders from non-responders. (**c**) Respective performances of the OT mini-classifier and of the RAS/RAF mutation status in predicting cetuximab sensitivity in the following cohorts OT-PDX (OT) (cross-validation), EPO PDX, the Gao et al.,[12] PDX (NV) and Khambata-Ford et al.,[46] primary tumours (KF). The mutation status was defined by mutations in codon 12 and 13 of KRAS or detected activating mutations in KRAS, BRAF or NRAS (BRAF mutations: V600E; KRAS/NRAS mutations: G12, G13, Q22, Q61, A146). In an additional setup, stable disease (SD) samples were excluded. The dot plot shows the specificity, sensitivity and balanced accuracy of the predictions. The 0.8 value is highlighted.with a black dotted line. (**d**) Same as in (**c**) but taking as validation cohort 164 samples merged from OT, EPO, NV and KF. (**e**) Distribution of the predicted responders and resistant patient tumours (tumour purity ≥ 40%) in the three molecular groups (ECM/EMT, ASCL2/MYC, Entero/Goblets).

signature shown in Fig. 9e, we built a mini-classifier of 16 genes able to segregate responders from non-responders, using a combination of recursive feature elimination and SVM algorithms (see Methods, Supplementary Data 17, Fig. 10a,b). We tested the stability of this mini-classifier by cross-validation on the OT xenografts, showing an overall improved performance over the *RAF/RAS* mutation status alone in predicting the outcome of cetuximab treatment, in terms of both sensitivity and specificity (Fig. 10c). In addition, we validated the classifier's predictive power on two independent CRC xenograft cohorts and one independent human cohort. We generated RNAseq data on a previously reported collection from EPO ($n = 60$)[58] and downloaded the informative data from Novartis PDXs (NV, $n = 36$)[12], and from a human cohort of 68 patients KF ($n = 68$)[46], all treated with cetuximab (Fig. 10c; Supplementary Table 1). On the xenograft cohorts, the OT mini-classifier outperformed the mutation status-based predictor for predicting responders and achieved a sensitivity of 0.92 and 1 for *KRAS/NRAS/BRAF* wild type samples on the NV and EPO xenografts, respectively.

In the Khambata–Ford (KF) cohort, the *KRAS* mutation status was highly sensitive but not very specific, because of a number of WT-*KRAS* tumours that did not respond to cetuximab (of note, the *BRAF* status was not tested in this retrospective study), whereas the OT mini-classifier exhibited higher sensitivity (0.83), and specificity (0.86) (without stable disease). Overall, the OT mini-classifier outperformed the *BRAF/KRAS/NRAS* status in predicting cetuximab response, despite inherent differences between datasets including patients from different populations and ethnic origins (US, European and Chinese), tumour stages and methodologies (microarrays versus RNAseq). We then merged the EPO, NV and KF cohorts in a superset of 164 samples for testing the OT mini-classifier, showing a sensitivity of 0.94 and a specificity of 0.79 for the *KRAS/NRAS/BRAF* wild type samples (Fig. 10d; Supplementary Table 2). On the basis of this performance, we applied the OT mini-classifier to the original OT-patient cohort (Fig. 10e), predicting that sensitive cases were mainly found in the ASCL2/MYC group, in particular early stage tumours (Fisher's exact test $P = 0.0068$). Although a few sensitive tumours belonged to the ECM/EMT group, those were showing similarities with the ASCL2/MYC samples (Supplementary Fig. 8f, subgroup A).

## Discussion

We report here a unique CRC biobank comprising 59 xenografts and 35 organoids tested on a panel of clinically relevant compounds. This is the first large study providing extensive molecular characterization of a patient donor cohort, encompassing all the disease stages, and representative of the typical CRC molecular groups, with matched *in vivo* and *in vitro* models.

A key issue for evaluating the potential of pre-clinical models relies on their capacity to retain the complex molecular and biological characteristics of the parental tumours. We showed that most models recapitulated the genetic landscapes of donor tumours, whereas clonal discordances found at early passages were attributed to ITH, reflecting the initial sampling of the bulk tumour. We identified few discordant mutations potentially impacting treatment outcome, (for example, *EGFR*, *PTCH1*), suggesting that heterogeneous parts of the tumours could respond differently to targeted drugs. This issue needs to be taken into consideration in CRC personalized settings, requiring ideally the establishment of multiple models recapitulating ITH, challenging the $1 \times 1 \times 1$ scheme[12]. Transcriptome landscapes indicated that cellular and biological pathways were reasonably well conserved in both model types, except for immune and stromal environment. Xenografts are currently considered as gold

standard tools for precision medicine, whereas the potential of organoid technologies is in early evaluation[16]. We found that PDXs appeared closer to the CRC molecular groups than PDOs, in particular those derived from the ASCL2/MYC group, but follow-up studies are required for confirming this observation. PDOs featured higher expression of genes involved in xenobiotic and fatty acid processes, possibly impacting cancer cell survival[34], and may exhibit higher plasticity, displaying heterogeneous stem/differentiated enterocyte cell populations. These differences have a crucial impact on the choice of a suitable *ex vivo* model system for compound screening, and were investigated by comparative analyses of drug response in genetically identical PDO/PDX sibling pairs. Striking sensitivity differences between the model systems were seen for AZD8931 more potent in PDOs than in PDXs. Evaluation of drug response depends on pharmacokinetic parameters such as the concentration of active compounds at the tumour site. Assessing the clinical relevance of potency levels *in vitro* should consider clinical achievable drug plasma concentration ($C_{max}$ or $C_{ss}$, maximum or steady-state concentrations), providing these values are known from clinical studies. Drugs exhibiting low potency in cell models, yet having high tolerable $C_{max}$ values in patients might trigger a response although PDOs were scored insensitive. For instance, regorafenib reaches a $C_{max}$ value of $2.5 \, \mu g \, ml^{-1}$ following a single 160 mg dose in patients with advanced solid tumours[59].

Here, we tested 11 drugs in 59 PDX models recapitulating most of the molecular features of the donor human tumours, providing the equivalent of phase II/III-like response data. Dissecting the stromal human components in models, leading to more specific signatures of the tumour cells, contributed to identify novel classifiers of response to the clinical compounds 5-FU and cetuximab, correlating responders with specific CRC molecular sub-groups. We derived a novel classifier predicting cetuximab response with high sensitivity and specificity, outperforming current genetic biomarkers and able to predict the drug sensitivity of *RAS/RAF* wild-type tumours, a group of tumours, which up to now lacked efficient biomarkers associated with EGFR blockade sensitivity.

We demonstrated here the power of integrative analyses for capturing the complexity of the CRC biology. Understanding the molecular make-up of each tumour and model is a paradigm becoming realizable with increasingly affordable NGS protocols and availability of powerful analysis tools. We identified clinically relevant gene fusions, and provided the first CRC *in vivo* model of *ALK* fusion responding to crizotinib. The OncoTrack study provides a large biobank and data repository based on patients and pre-clinical models with unprecedented breadth and depth, and a compendium of drug sensitivity data, a resource that can be exploited further for improved drug discovery and understanding of CRC biology.

## Methods

**Patients.** In this study we included patients with colorectal cancer aged between 18 and 100 years. The only exclusion criterium has been infectious diseases. Samples were stored according to the current GCP guidelines. Informed consent was obtained from all human subjects included in the study. The study was approved by the local Institutional Review Board of Charité University Medicine (Charité Ethics Cie: Charitéplatz 1, 10117 Berlin, Germany) (EA 1/069/11) and the ethics committee of the Medical University of Graz (Ethic commission of the Medical University of Graz, Auenbruggerplatz 2, 8036 Graz, Austria), confirmed by the ethics committee of the St John of God Hospital Graz (23-015 ex 10/11).

**Nucleic acid preparations.** Nucleic acid preparations were performed either using the AllPrep DNA/RNA/Protein Mini Kit (Qiagen, 80004) or the AllPrep DNA/RNA/miRNA Universal Kit (Qiagen, 80224). DNA extraction from blood was carried out using the QIAamp DNA Blood Maxi Kit 10 (Qiagen, 51192). Concentrations were determined on Qubit Fluorometer. RNA integrity was evaluated with Bioanalyzer 2100 (Agilent, Palo Alto, CA).

**Targeted sequencing.** Targeted sequencing libraries were prepared with the TruSeq Custom Amplicon Kit (Illumina, FC-130-1001) and Index Kit (Illumina, FC-130-1003) following the True Seq Custom Amplicon Low Input Library Prep protocol (October 2015). TruSeq Custom Amplicon panels were designed with Illuminas DesignStudio. Paired-end (PE) libraries were sequenced on Miseq PE 151 dual Index.

**Mutation validation with Sanger sequencing.** PCR primers are listed in Supplementary Table 3. PCR products were purified and processed by Sanger sequencing (Eurofins MWG Operon).

**Microsatellite status.** Microsatellite status was analysed using the five mono-morphic markers BAT25, BAT26, NR21, NR24 and NR27. Pentaplex PCR reactions were performed using the primers given in Supplementary Table 4. PCR reactions and capillary electrophoresis were performed by Eurofins Genomics GmbH (Germany).

**Whole genome sequencing.** High coverage WGS libraries were prepared with the TruSeq DNA Sample Prep v2 kit (Illumina, set A: FC-121-2001; set B: FC-121-2002) following the Illumina Low Throughput (LT) Protocol (August 2011). Paired-end libraries were sequenced on HiSeq 2,000/2,500 instruments with v3 chemistry using $2 \times 101$ bp reads to $\times 50$ coverage. For low coverage, WGS libraries were prepared using Nextera Rapid Capture Exome and Expanded Exome Kit (Illumina, FC-140-1006), but omitting the exome enrichment step. Paired-end libraries were sequenced ($2 \times 51$ bp) on HiSeq 2,000/2,500 instruments with v3 chemistry to $\times 1$ coverage.

**Whole exome sequencing.** WES was carried out on either SOLiD or Illumina platforms. SOLiD libraries were prepared either with Sure Select XT Human All Exon 50 MB (Agilent Technologies, 5190-0407) or with SureSelect Human All Exon V4 (Agilent Technologies, 5190-4631). Sequencing was carried out either on SOLiD 5500 in Frag75/ECC mode or on SOLiD Wildfire in Frag50/ECC runs using single-end mode. For Illumina, libraries were prepared with Nextera Rapid Capture Exome and Expanded Exome Kits (Illumina, FC-140-1006). Paired-end libraries ($2 \times 51$ bp) were sequenced on HiSeq 2000/2500 instruments with v3 chemistry.

**Whole transcriptome sequencing.** RNAseq libraries were prepared using either TruSeq RNA Sample Prep Kit v2 (Illumina, set A: RS-122-2001; set B: RS-122-2002) with modifications preserving strand-specific information[60] or TruSeq Stranded mRNA Sample Prep Kit (Illumina, set A: RS-122-2101; set B: RS-122-2102). For ten total RNA samples we used the Ribo-Zero Magnetic Gold Kit (Epicentre, MRZG12324). Sequencing ($2 \times 51$ bp) was performed on HiSeq 2000/2500 instruments with v3 chemistry.

**DNA data processing.** DNA reads were aligned to the human reference genome hg19 using BWA (bwa0.7.7-r441-mem for 75/101 bp, bwa0.5.9-r16-aln for 51 bp reads). For xenograft samples, the human and mouse DNA reads were deconvo-luted after mapping to references from human hg19 and mouse mm9 genome versions.

**Copy number variants.** Copy number variants were estimated using the BICSeq algorithm[61] and the read coverage data of tumour versus normal pairs. We inferred ploidy using the B allele frequencies of heterozygous germline variants. For low coverage WGS without matching blood data we used as a proxy an electronic pool of six sex-matched normal samples.

**Somatic SNVs.** Somatic SNVs were detected using established pipelines based on VarScan2 combined with RNAseq data and functional annotation of the variants based on Ensembl v.70. Somatic indels were detected using SAMtools and Dindel[62].

**RNA data processing.** RNA reads were aligned to hg19 using BWA and SAM-tools. Mapped reads were annotated using Ensembl v70. Gene expression levels were quantified in reads per kilobase of exon per million mapped reads (RPKM).

**Gene fusions.** Gene fusions were detected by RNAseq using deFuse (v0.5.0) and TopHat2-Fusion (v2.0.3). High-confidence events were selected and subjected to visual inspection. Fusion transcripts were annotated on Ensembl gene annotation v62. For validation, 50 ng of total RNAs were reverse transcribed and fused transcripts were amplified using the dART 1-Step RT-PCR Kit (EURx #E0803-02) using primers located upstream and downstream of the transcript breakpoints (Supplementary Data 4). RT-PCR products were purified and processed by capillary Sanger sequencing (Eurofins MWG Operon).

**SciClone.** Investigation of tumour clonality in corresponding patient, PDX, PDO samples of the same donor was performed using the program sciClone version 1.1 (ref. 24). As an input we utilized all SNVs scored in diploid regions. Clustering of the data were performed by the sciClone tool based on variant allele frequencies of all informative SNVs that have a minimum coverage of 48 reads for patient samples and 24 reads for PDX and PDO models. Data for corresponding patients, PDX, PDO were given simultaneously as input to the sciClone program. We obtained informative results for 41 out of 62 samples.

**Analysis of EPO cohort.** For 57 of the 60 PDXs mutation information from allele-specific RT-PCR (Custom TaqMan SNP Genotyping Assays, Applied Biosystems) for *KRAS* G12, G13 and A146, for *BRAF* V600E and for *PIK3CA* E542K, E545K and H1047R was available (ref. 58). Additionally, we generated RNAseq data and investigated for sequencing reads carrying mutations at *KRAS* codons 12,13,22,61,146, *BRAF* codon 600, *NRAS* codons 12,13,61, *PIK3CA* codons 542,545,1047,420,88. Gene fusion detection was carried out using Tophat2.

**Comparison of corresponding patient tumours and models.** For the comparison of SNVs/indels we applied a two-step analysis. At first, SNVs/indels were called in patient tumour, PDX and PDO samples as described above with adequate criteria. In a second step, for corresponding sample pairs or sample trios we rechecked the allele frequency of SNVs/indels detected in minimally one of the corresponding samples. SNVs/indels showing a minimum allele frequency of 2% were considered as present in a corresponding sample if not detected in the first step (Supplementary Data 3). A mutation was consider damaging, if either Poly-Phen $> 0.7$, MutationTaster $> 0.7$ or SIFT $< 0.05$. Unfiltered fusion candidates detected by Tophat2 (fusions.out) and/or deFuse (results.tsv) were mutually checked and compared between corresponding samples (Supplementary Data 4). Copy-number variants were estimated as described above and compared between corresponding samples.

*Cancer relevant gene selection.* Cancer relevant gene selection was done by taking the overlap with 31 significantly mutated genes in the CRC TCGA study[21] and 86 genes recurrently mutated in CRC from the TCGA pan-cancer analysis[25].

**Identification of molecular tumour groups.** RPKM values of 65 primary CRC samples (purity $\geq 40\%$) were analysed with non-negative matrix factorization NMF and the CLICK algorithms, respectively. For both methods, we selected the most variable genes (3,529 genes, cut-off $> 0.5$ RPKM in more than three samples) in the 65 primary tumour samples using quantile absolute deviation $(QAD = P_{75}(|Xi - median(X)|)$, calculated on raw and $\log_2$ transformed RPKMs.

**Non-negative matrix factorization.** Cluster stability was assessed by applying the factorization step 60 times, leading to three tumour classes based on the cophenetic score. Corresponding gene signatures were identified by differential gene expression analysis (DGEA) using the R package edgeR (v.3.6.0). Genes were filtered by $|\log_2(FC)| \geq \log_2(1.7)$ (fold change), FDR $\leq 0.01$ and difference of mean expression $\geq 1$. Among these, genes up-regulated in one or more of the NMF groups were selected, resulting in 961, 20 and 146 specifying OT_NMF1, OT_NMF2 and OT_NMF3, respectively (Supplementary Data 10).

**CLICK clusters.** RPKM values were $\log_2$ transformed and centred by the trimmed mean. For gene co-expression clusters, best results were achieved with an expected homogeneity value of 0.60 leading to a separation of $-0.085$, to an overall homogeneity of 0.753 and 13 clusters (named OT_C1 - OT_C13). A 14th cluster was excluded since it mainly contained uncharacterized genes. On the basis of the mean pattern obtained for each CLICK cluster, tumour samples were divided into two groups with partitioning around medoids (pam) implemented in R[63]. We selected differentially expressed genes with $|\log_2(FC)| \geq \log_2(2)$, FDR $\leq 0.01$ and difference of mean expression $> 1$, resulting in 838, 144, 127, 92, 101, 74, 58, 54, 52, 49, 61, 42 and 49 genes in CLICK clusters 1–13, respectively (Supplementary Data 10).

**Mean pattern matrix of primary CRCs.** In addition to the OT 16 gene signatures (OT_NMF1, OT_NMF2, OT_NMF3 and OT_C1-13), we collected signatures from external resources: (1) 286 inflammation genes extracted from: expression arrays 'Qiagen Human Cancer Inflammation & Immunity Crosstalk RT2 Profiler PCR Array', 'Qiagen Inflammatory Cytokines and Receptors RT2 Profiler PCR Array', 'Qiagen Inflammatory Response and Autoimmunity RT2 Profiler PCR Array' and genes in the Gene ontology category 'inflammatory response' (Supplementary Data 10), (2) 19 published CRC groups signatures collected from 4 publications[1,3–5]. For Sadanandam *et al.*[4] and De Sousa *et al.*[1] gene signatures, we used the provided single list of classifier genes and assigned them to the different subtypes using centroids and PAM. From Schlicker *et al.*[3] we used the provided gene signatures. For Marisa *et al.*[3] we assigned the probes to the 6 subtypes by taking a $|\log_2(FC)| \geq 0.5$ and $P < 0.0001$ as cutoff on their list of 1,108 probes. We performed the conversion mapping of Affy-133-plus-2-probeset IDs to Ensembl IDs (v70) using BioMart (Ensembl),

obtaining a total of 38 signatures that we combined in a so-called 'mean pattern matrix' integrating 70 primary CRCs and 20 metastatic samples (tumour purity $\geq 40\%$). We calculated the mean patterns for each specific signature in each of the CRC samples (trimmed mean of $\log_2$ and Z-score transformed RPKM values) resulting in a matrix of $90 \times 38$ entries (the mean pattern values are shown in Supplementary Data 12). To test the stability of hierarchical clustering the approach pvclust (R package, v 2.0) was applied with 5,000 iterations and a sample size of 80, 85, 90, 95, 100, 105, 110, 115 and 120% (ref. 64).

**Clustering of the mean pattern matrix.** On the basis of the values of the mean pattern matrix, patient tumours were clustered into subgroups using the mclust package in R, which tested for 2 up to 15 expected clusters[28]. On the basis of the Bayesian information criterion (BIC), 3 and 6 clusters were the best solutions.

**Mean pattern matrix comparing CRCs with experimental models.** Mean RPKM values for each model and gene signature were calculated as described above. To compare the models directly to the initial patient samples, we used the mean and standard deviation of a gene in the patient tumour cohort and calculated the mean of the Z-transformed RPKM values.

**NMF groups in models.** NMF was applied to the most variable expressed genes based on the QAD (see above) resulting in 5,150 and 4,629 genes for PDX and PDO models, respectively. On the basis of cophenetic scores we obtained two clusters for the PDX and PDO samples, respectively. Differentially expressed genes between the two NMF groups were identified using edgeR (Supplementary Data 10). The attribution of NMF groups for each of the PDX and PDO samples is given in Supplementary Data 9.

**Single sample Gene Set Enrichment Analysis.** Single sample Gene Set Enrichment Analysis (ssGSEA) was performed using GSVA package in R[65]. We used as input $\log_2$ transformed RPKM values and gene sets of immune[66–68] and 16 OT signatures as referred in the text (OT_NMF1, OT_NMF2, OT_NMF3 and Click1-13). The homogeneity of variance was fulfilled for OT_C2, OT_C7, OT_C12, inflammation, OT_C5, OT_C13, OT_C8. The three main CRC groups were compared using one-way analysis of variance and Tukey's range test. For the pairwise comparison of the sub-groups and the main groups against MSI samples a Welch's $t$-test was applied. The $P$ values were adjusted for multiple testing using the Benjamini–Hochberg procedure (false discovery rate—FDR). Differences between groups of patients were considered significant with a FDR $\leq 0.01$ for the 16 OT signatures and FDR $\leq 0.05$ for immune signatures.

**Functional annotations.** Functional annotations were achieved by gene set over-representation analysis using the GePS Genomatix software (v3.10124 and v3.51106).

**Global gene expression profile comparison.** Global gene expression profile comparison of tumours and model systems was carried out using 90 original patient samples, as well as untreated controls of 54 PDX and 33 PDO (Fig. 5b). From the five xenografts that derived from 150_MET1 only the sample 150_MET1_XEN1 was included into the analysis, since all of them shared a similar expression profile. We performed a DGEA using edgeR and compared PDX versus patients, PDO versus patients and PDO versus PDX. To identify global differences in gene expression we applied strict cutoffs: FDR $\leq 0.005$, $|\log_2(FC)| \geq \log_2(2.5)$ and difference of mean expression $\geq 2$. Functional analyses of the up- and down-regulated genes were performed using the GePS Genomatix software.

**Pairwise gene expression profile comparison of models.** Pairwise gene expression profile comparison of models was applied on 17 corresponding PDO-PDX pairs using a pre-ranked gene set enrichment analysis (GSEA, Broad Institute) (Fig. 5a). To establish the gene ranking based on the expression values for a matching PDX $\mathbf{X}$ and PDO $\mathbf{Y}$, the ranking score $r$ was calculated by taking the geometric mean of the fold change and difference of expression,

$$c_{pg} = \log_2\left((x_{pg}+1)/(y_{pg}+1)\right)$$

$$d_{pg} = \log_{10}\left(|x_{pg}-y_{pg}|+1\right)$$

$$r_{pg} = \sqrt[2]{|c_{pg} \cdot d_{pg}|} \cdot \mathrm{sign}(c_{pg})$$

where $g$ indicates a specific protein coding gene and $p$ a specific pair of corresponding PDO and PDX. Gene sets were downloaded from ConsensusPathDB[69] (release 31, downloaded gene sets: Wikipathways, PID, Reactome, NetPath, KEGG) and MSigDB (downloaded gene sets: hallmark, http://software.broadinstitute.org/gsea/msigdb, v5.1). GSEA was performed using a java implementation of GSEA (v 2.220) with 3,000 gene permutations and classical statistics. A FDR $< 0.005$ was considered as significant.

**CMS classification.** 90 OT CRC samples (tumour purity $\geq 40\%$) were classified into the consensus molecular subtypes (CMS) reported by Guinney et al.[2] We applied the provided single-sample predictor (SSP) and random forest (RF) CMS classifier to the 90 OT CRC samples (R package downloaded from www.synapse.org/#!Synapse:syn2623706/wiki/). With the default parameters, 25/90 and 24/90 of the samples (28%) were unclassified using RF and SSP, respectively. Thus, we lowered thresholds for subgroup assignments (RF: minPosterior = 0.4; SSP: minCor = 0.1, minDelta = 0.03). For RF we used a minimal posterior probability of 0.4 since it shows just a slightly lower specificity compared with the default value 0.5 (Guinney et al.[2]; Supplementary Fig. 3). Applying RF and SSP with lowered thresholds 8 and 17 samples remained unclassified, respectively (Supplementary Data 9). The lowered thresholds were also applied on the PDX and PDO OT cohorts.

**CCLE 2D cell lines.** BAM-files for 41 COAD CCLE cell lines (April 2014) were downloaded from https://cghub.ucsc.edu/index. Gene expression values were normalized as RPKM (see methods RNA data processing) using gene models from Ensembl Release 73.

**Drug response analysis preprocessing.** Five xenografts that derived from one CRC (150_MET1) shared highly similar global expression profiles and were merged into one artificial single sample by taking an average of the RPKM and of the T/C values, respectively. Four PDX (234, 372, 283 and 128) had no RNAseq data. For cetuximab one PDX (261) was considered as outlier and excluded from the analysis, since it was the only sample that showed a T/C value far above 100%. Two PDOs (159,161) had no RNAseq data.

**Correlation of drug sensitivity values.** Drugs were pairwise and per model system compared based on $\log_{10}$ normalized $IC_{50}$ values for 35 PDOs and T/C values for 53 PDXs and by calculating Spearman's rank correlation coefficient.

**Drug response gene signatures in PDX and PDO.** We performed differential gene expression (DGE) analysis using the R package edgeR per drug and model system to identify signatures associated with drug response results based on the four response categories: strong, moderate, minor, resistant (see below drug response). DGE analysis was applied in different setups as follows: (a) combined strong + moderate versus combined minor + resistant, (b) combined strong + moderate versus minor versus resistant, and (c): 20 most sensitive versus 20 least sensitive PDX or 10 most sensitive versus 10 least sensitive PDO. Genes were filtered by FDR $\leq 0.01$, $|\log_2(FC)| \geq 1$ and RPKM difference $\geq 1$. In addition, (setup d), we used the $IC_{50}$ or T/C values as phenotype vector in a general linear model (GLM) provided by the edgeR package. Genes were filtered by FDR $\leq 0.01$ and dispersion $< 4$. Gene signatures associated with a given drug response were generated by combining results from setups a to d. Low expressed genes were filtered by an expression $\geq 1$ RPKM in minimally five PDX or three PDO samples and by a mean expression $\geq 0.8$ RPKM for 5-FU (PDX), Avastin (PDX), vandetanib (PDO), afatinib (PDX and PDO), AZD8931 (PDX and PDO), gefitinib (PDO) and cetuximab (PDX) (Supplementary Data 15).

**Building drug response classifiers for cetuximab and 5-FU.** Of the 241 genes making the cetuximab signature only 179 genes could be mapped to external datasets[12,46,58]. The following procedure was applied to the 179 mapped genes. RPKM values were $\log_2$ transformed and $z$-score normalized. Eleven genes that showed lower mean expression between highly correlated gene pairs (Pearson correlation $\geq 0.8$) were excluded in two iterations. For the drug response classifier a linear support vector machine (SVM) implemented in the R package 'e1071' (v1.6-7) was trained on 48 PDX of the OT cohort (14 responding and 34 resistant PDX). To address the imbalance of the training set, a class weighted SVM was used and the hyperparameter $C$ was tuned for each of classes resistance and response ($C_{resis}$, $C_{resp}$). The feature (gene) selection included feature ranking and feature size selection. To avoid overfitting of the SVM, a SVM recursive feature elimination (SVM-RFE) was used for feature ranking, similar to the approach of Duan et al.[70] In each recursive step of our adaptation of the procedure, the hyperparameter $C_{resis}$ and $C_{resp}$ were tuned via grid search with a stratified bootstrap (100 iterations), the ranking scores were calculated based on a stratified leave-n-out resampling (200 iterations). The performance of a hyperparameter set was evaluated using the $F_1$-score. The calculation of the ranking score was based on the weight vector $w$ of a linear SVM and not $w^2$ as described[70]. For the cetuximab classifier the parameter set with the third highest $F_1$-score was taken as optimal solution, since it showed the highest sensitivity among the top three: $C_{resis} = 0.05$, $C_{resp} = 0.3$, feature size = 16. The described procedure resulted for the 5-FU gene signature in a mini-classifier with 14 genes. The parameter set with the eleven highest $F_1$-score was taken as optimal solution, since it showed the highest sensitivity among the top results: $C_{resis} = 0.007$, $C_{resp} = 0.02$.

*Validation of the cetuximab response classifier.* The cetuximab response classifier (16 genes) was validated on the OT PDX cohort, on one external human cohort for 80 metastatic CRC patients with array expression (Affymetrix U133A v2.0 GeneChips; Khambata-Ford et al.[46]) and two external PDX cohorts with 59

and 60 models with RNASeq data (Gao et al.[12], Pechanska et al.[58]) all treated with cetuximab. For 68 samples of the Khambata-Ford and for 36 samples of the Gao dataset expression and cetuximab response data were available. Ensembl gene identifiers were mapped to u133av2 probeset IDs (Khambata-Ford) and to gene symbols (Gao). The expression values of the external PDX cohorts were log$_2$-normalized and of all three external data set were z-score transformed. Four response categories were given for the Gao and Khambata-Ford data set: complete response (CR), partial response (PR), stable disease (SD) and progressive disease (PD). PDX of the EPO were divided in four response categories based on given T/C values as described in this paper. The performance of the classifier was estimated from the number of true positive (TP), false positive (FP), true negative (TN) and false negative (FN) predictions as well as the sensitivity, specificity and balanced accuracy. Cross-validation on the OT PDX cohort was achieved via a 100 times repeated 10-fold cross-validation. Performance values were averaged over the repeats. In a second analysis, SD samples were excluded to determine their influence on the classifier's performance (KF: 49 samples, Gao: 32 samples). Additionally, we tested the performance of the classifier on KRAS wild type and all-RAS/RAF wild type samples. For the KRAS wild type, mutations in codon 12 and 13 of KRAS were considered. For the all-RAS/RAF wild type, KRAS and NRAS mutations (G12, G13, Q22, Q61, A146), as well as BRAF mutations (V600E) were checked. The Khambata-Ford data set provided only mutations in codon 12 and 13 of KRAS.

*Cross-validation of 5-FU response.* The 5-FU mini-classifier (14 genes) was cross-validated on the OT PDX cohort via a 100 times repeated 10-fold cross-validation. Performance values were averaged over the repeats. The performance of the classifier was estimated from the number of true positive (TP), false positive (FP), true negative (TN) and false negative (FN) predictions as well as the sensitivity, specificity and balanced accuracy.

**Establishment of PDO cell cultures.** Upon resected sample receipt, fatty and necrotic tissues were removed macroscopically. Remaining tissue was rinsed with HBSS (Gibco), minced, and digested by Collagenase IV (Sigma-Aldrich), DNaseI (AppliChem, Germany) and Dispase (StemCell Technologies, Germany) at 37 °C for 60 min, followed by pelleting the suspension at 300 $g$ for 3 min, re-suspension in medium and filtration steps as in Konno et al.[71]. The 40–100 μm aggregates were centrifuged at 300 $g$ for 3 min. After depletion of red blood cells using Red Blood Cell Lysis Solution (Miltenyi, Germany), cells were mixed with phenol-red free growth factor-reduced Matrigel (Corning) and seeded into 24-well plates. Solidified droplets were carefully overlaid with 500 μl of culture medium as in Sato et al.[72]. During the first week 1.25 μg ml$^{-1}$ Amphotericin B and 10 μM of the ROCK-II inhibitor Y27632 (Sigma-Aldrich) were added to cultures. The cultures were passaged when the aggregates reached a diameter of approximately 800 μm. Cellular aggregates were released from Matrigel by adding 5 ml Advanced DMEM/F12 followed by centrifugation. Pellets were digested with TrypLE (Gibco). Trypsinization was stopped with 5 ml Advanced DMEM/F12 and cell clusters were re-plated on a 24-well plate. PDO cell cultures were generated from 41 patient tumours and 5 xenografts (Supplementary Data 1). The cell cultures were routinely tested for Mycoplasma contamination and found to be negative.

For immunohistochemistry, 2 μm de-paraffinized FFPE tissue sections of donor tumours or PDO cultures grown for five days were stained using the primary antibodies anti-CK7 (clone OV-TL12/30, Dako, Germany), anti-CK20 (clone KS20.8, Dako), anti-CDX2 (clone CDX2-88, BioGenex, USA) and anti-KI67 (clone MIB-1, Dako) for 32 min at 37 °C, ultraView DAB detection kit (Ventana, USA) on the BenchMark XT instrument (Ventana). Counterstaining was performed with Hematoxylin II Counterstain and Blueing Reagent (Ventana) for 4 min. For immunofluorescence imaging, PDO cell aggregates were fixed and permeabilized with 4% PFA/1% Triton X for 30 min, followed by treatment with 1% Triton X overnight at 4 °C. PDO aggregates were then washed in PBS with 10% FCS. Primary anti-Ezrin antibody (clone 3C12, Thermo Scientific) was incubated at 4 °C for 48 h and removed by washing in PBS with 10% FCS. Secondary antibody (Alexa Fluor488, Invitrogen) was added at 4 °C overnight and removed by washing in PBS. Nuclei were stained with DAPI (Sigma-Aldrich) for 30 min. F-actin was stained accordingly with TRITC-labelled Phalloidin (Sigma-Aldrich). Microscopy was performed with a Zeiss Axiovert 400 microscope (Zeiss, Germany).

**Semi-automated high-throughput drug sensitivity assays.** PDO cultures were digested with TrypLE (Gibco) to single cell suspension. Trypsinization was stopped with Advanced DMEM/F12. We seeded 5,000 cells/well in growth factor-reduced Matrigel into 384-well plates using a robotic platform (Tecan, Spain). Cells were cultured for four days before compound treatment. Growth curves were determined by assaying the cell viability by luminescence (CellTiter-Glo, Promega), using the EnVision plate reader (PerkinElmer) 30 min after the addition of the reagents. The 384-well plate layout included appropriate Min (minimum signal, 5 μM staurosporine) and Max (maximum signal, vehicle, 0.25% DMSO) controls to determine signal intensity cutoffs. PDO cultures were screened in two replicates with the test compounds ranging from 60 μM to 3.05 nM with 1:3 serial dilution steps, and cetuximab ranging from 5 μg per ml to 0.25 ng per ml with 1:3 dilution steps (with maximum signal wells containing medium only). The treatment duration covered two population doubling times for each cell culture strain. Plate uniformity was validated as previously described[73] and in accordance

with published Eli Lilly-NIH Chemical Genomics Center Guidelines for assay enablement and statistical validation[74]. $E_{max}$ values were calculated as the percentage of inhibition at the maximum included concentration. Relative IC$_{50}$ values were calculated with the four-parameter nonlinear logistic equation and were classified into four response categories based on the tested concentration range: resistant ($>5.0656$ μM) and minor (from $5.0656$ μM to $0.4277$ μM), moderate (from $0.4277$ μM to $0.0361$ μM) or strong responders ($\leq 0.0361$ μM). For cetuximab the four response categories are based on the log(IC$_{50}$) values and are defined as: resistant ($>-1.483$ μM) and minor (from $-1.483$ μM to $-3.63$ μM), moderate (from $-3.63$ μM to $-4.703$ μM) or strong responders ($\leq -5.777$ μM).

**Live imaging and confocal microscopy.** For the time-lapse analysis, the organoid growth in 384-well plates was monitored using a HC PL APO $\times 10/0.40$ AN ($\times 10$) objective, a Hamamatsu ORCA-AG CCD camera and an inverted motorized microscope (Leica DMI 6000B) coupled with an incubation system to control the temperature and CO$_2$ levels during the experiments. Images were taken every 15 min for 72 h using the Leica LAS AF software (Version 2.4.1). Confocal microscopy was carried out using a Leica TCS SP5 X confocal microscope equipped with a resonant scanner, a dry $\times 20$ Plan Apochromatic, 0.7 AN objective, and Leica LAS AF software (Version 2.4.1) for image capturing, and the Imaris software (Bitplane) for image analysis.

**WNT reporter assay.** Wnt pathway activity was assessed by lentiviral transduction with the Cignal Lenti TCF/LEF Reporter (GFP) Kit CLS-018G (QIAGEN, Hilden, Germany). Organoids were released from Matrigel, plated into 96well round bottom ultra-low attachment plates (Corning) with 20 μl virus suspension. Following transduction, organoids were re-plated in Matrigel and selected with puromycin. After selection, organoids were released from Matrigel, digested to a single cell suspension and sorted into GFP-high/GFP-low/GFP-negative fractions by FACS. Ribonucleic acid was isolated from the sorted cells using the AllPrep DNA/RNA Mini Kit (QIAGEN) and provided for RNA-Seq.

**Development and characterization of PDXs.** Resected tumour tissues were transplanted to immunodeficient mice (NMRI nude or NOG, Taconic, Bomholdtgard, DK- Tac:NMRI-Foxn1nu, females, 6–8 weeks at start of transplantation) using previously described methods by Fichtner et al.[75]. Animal experiments were carried out in accordance with the United Kingdom Coordinating Committee on Cancer Research regulations for the Welfare of Animals and of the German Animal Protection Law and approved by the local responsible authorities. EPO strictly follows the EU guideline European convention for the protection of vertebrate animals used for experimental and other scientific purposes. (EST 123)' and 'German Animal Protection law -Version July 2014' (Tierschutzgesetz: zuletzt geändert durch Art. 3 G v. 28.7.2014 I 1308). Further we handle our animals according to Regulation on the protection of experimental scientific purposes or other Purposes used animals (Tierschutz-Versuchstierverordnung- TierSchVersV: Geändert durch Art. 6 V v. 12.12.2013 I 4145). Compliance with the above rules and regulations is monitored by the Landesamt für Gesundheit und Soziales (LAGeSo) which is the responsible regulatory authority monitoring the animal husbandry based on the German Animal Welfare Act, last revised in 2014. Approval was given after careful inspection of the site including bedding, feeding & water, ventilation, temperature & humidity, cleaning and hygiene concepts. Mice were monitored three times weekly for tumour engraftment for up to 3 month. Engrafted tumours at a size of about 1 cm$^3$ were surgically excised and smaller fragments re-transplanted to naive NMRI nu/nu mice for further passage. Within passage 1–3 numerous samples were cryo-conserved (DMSO-medium) for further experiments. Tumours were passaged not more than 6 times. For confirmation of tumour histology, tumour tissue was formalin fixed and paraffin embedded (FFPE) and 5 μm sections were prepared. Samples were stained according to a standard protocol for hematoxilin, eosin and Ki67 to ensure xenograft comparability to the original specimen. Cases with changed histological pattern were sent for pathological review and outgrowth of lymphoproliferative disorders was excluded. In this study, no blinding was done.

*In vivo* **drug response testing of the xenografts.** Response to the selected compounds was evaluated in early passages using the design of a preclinical phase II study. Tumour fragments of similar size were transplanted subcutaneously to a large cohort of mice. At palpable tumour size (50–200 mm$^3$), mice were randomized to treatment or control groups consisting of 5–6 animals each. Doses and schedules were chosen according to previous experience in animal experiments and represent maximum tolerated or efficient doses. Applied schedules are shown in Supplementary Table 5. The injection volume was 0.1–0.2 ml per 20 g body weight. Treatment was continued over a period of four weeks (4 cycles) or till tumour size exceeded 1 cm$^3$ or animals showed loss of >15% body weight. From the first treatment day onwards the tumour volumes and body weights were recorded twice weekly. At the end of the treatment period animals were sacrificed, blood and tumour samples collected, and stored in liquid nitrogen immediately.

Animal welfare was controlled twice daily. Tumour volume was calculated from the length and width of subcutaneous tumours ($V = (\text{length} \times (\text{width})^2)/2$). Sensitivity to the tested compounds was determined as tumour growth inhibition

by treatment in comparison to the control (T/C) on each measurement point. Efficacy of the tested drugs in PDX models was classified by end-point T/C (treated/control) values expressing tumour growth delay of treated versus untreated (control) mice, with the following categories: T/C ≤ 10% as strong tumour growth delay, T/C 11–25% as moderate tumour growth delay, T/C 26–50% as minor tumour growth delay, and T/C > 50% as resistant. Tumours with a T/C < 25% can be considered to represent sensitivity in terms of (partial) tumour regression or stable disease.

For comparison, treatment response was in parallel evaluated using the adopted, stringent clinical response criteria (RECIST)[76]. We calculated the relative tumour volume (RTV) as the ratio of the tumour volume at the end of treatment/ tumour volume at the start of treatment.

The revised clinical response (RECIST) criteria taking as reference the baseline sum diameters define:

- Complete Response (CR): Disappearance of all target lesions. RTV = 0
- Partial Response (PR): At least a 30% decrease in the sum of diameters of target lesions (RTV < 0.7)—Progressive Disease (PD): At least a 20% increase in the sum of diameters of target lesions. (RTV > 1.2)
- Stable Disease (SD): Neither sufficient shrinkage to qualify for PR nor sufficient increase to qualify for PD. (RTV 0.7–1.2). As T/C and RTV are condensed summary parameter, no standard deviation values for replicate measurements are given in the Supplementary Data 14—these values have been determined and are available in the raw data.

**DNA methylation profiling.** 500 ng genomic DNA was bisulfite converted (EZ DNA Methylation Kit, Zymo Research) in accordance with the manufacturer's protocol with alternative incubation conditions (that is, 16 × cycles (95 °C for 30 s, 50 °C for 60 min)). Following bisulfite conversion, 14 randomly selected samples were quality controlled with qPCR using primers designed to anneal to genomic and bisulfite converted DNA (Primer are listed in Supplementary Table 6). Bisulfite converted DNA extracts showing ΔCt > = 5 between genomic and bisulfite converted primer pairs were hybridized to Infinium 450 K BeadChips (Illumina) and scanned with iScan (Illumina). Raw data (IDAT files) were pre-processed using *minfi* implemented in R. DNA extracts with more than 5% low detection metrics ($P < 0.05$) were excluded. DNA extracts with bisulfite conversion efficiencies[77] < 95% and probes mapping to chromosomes X and Y were excluded. Following DNA extract- and probe- triage, we subjected non-normalized[78] methylation values (β, the methylated fraction of cells assayed) to Singular Value Decomposition analysis[79], which revealed a significant batch effect that was overcome by normalizing raw intensity levels using functional normalization within *minfi*, according to a subsequent SVD analysis. For the purposes of counting 'methylated' probes, we defined methylated probes as having β > 0.3. We selected the top 5% most variable probes ($n = 22,358$) using the primary colon cancer samples, exclusively. This corresponded to a standard deviation across beta values greater than 0.19. This probe set was then used to perform hierarchical clustering (distance = 'Euclidean', linkage = 'complete') of primary and metastatic samples.

**Data availability.** The complete set of NGS data for patient tumours, matching reference blood samples, xenografts and cell models are available upon request in the European Genome-phenome Archive (EGA) of the EBI data repository under Accession number EGAS00001001752. The list of established CRC PDO and PDX models is implemented in the EPO website (www.epo-berlin.com). Academic groups and industrial companies can have access to the PDX models at EPO and to the PDO models at the biobank of the Charité Comprehensive Cancer Center (https://cccc.charite.de).

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

## Acknowledgements

The research leading to these results has received support from the Innovative Medicines Initiative Joint Undertaking under grant agreement no. 115234 (OncoTrack), resources of which are composed of financial contributions from the European Union's Seventh Framework Programme (FP7/2007–2013) and EFPIA companies' in-kind contribution (www.imi.europa.eu). We thank the NGS technical team of the Department of Vertebrate Genomics/OWL Gene regulation and Systems Biology of Cancer (Daniela Balzereit, Simon Dökel, Matthias Linser, Alexander Kovacsovics) and of the Alacris Theranostics GmbH (Marcus Albrecht, Anna Kosiura, Sabine Schrinner, Jeannine Wilde) for their excellent work in generating the NGS data used in this study and Sabine Thamm for the validation experiments. We also thank Stephanie Staudte, Dorothea Przybilla, Cathrin Davies, Maria Rivera, Katharina Scholl for technical assistance, and the MPIMG core IT group for its support. We acknowledge Mathieu Boniol, Hannes-Friedrich Ulbrich and Ton Coolen for their expertise and discussions on statistical analysis, Andreas Dahl, David Meierhofer, Niko Hildebrandt, Ivonne Marondel, Anthony Rowe, Ben Sidders, for useful discussions, Carl Steinbeisser for project management support. We also thank Laura Álvaro, Ester Arroba, Eduardo Goicoechea and Julia Gutiérrez from Eli Lilly and Company for technical assistance. We thank Amin El-Heliebi, Tabea Hohensee, Andreas Punschart, Philipp Stiegler from the Graz Hospital for helping in collecting clinical samples. Development and characterization of the PDO models were supported by DKTK (RSC) and Berliner Krebsgesellschaft (CRAR; Grant# 201402). This work was supported in part by the Max Planck Society, by the Swedish Research Council, and VINNOVA.

## Author contributions

The clinical cohort was managed, recruited and tissue samples were distributed to other partners with the help of C.S., N.G.S., S. Lax, S.U., I.K., A.F., S. Liebs under the leadership of J. Haybaeck and U.K. NGS data generation and analysis: M. Sultan, M.-L.Y. and T.B. supervised the operations in the sequencing platforms and analyses were performed under the leadership of M.L.Y. Data processing and genome analyses were carried out by M. Schütte, N.A.A., C.L.W., V.A. Transcriptome analyses were carried out by T.R. and C.J. Gene fusion were analysed by H.J.W., M. Schütte, T.R. and N.A.A. Methylation analysis was provided by L.M.B. and J.E.B. under the leadership of S.B. Experimental models: PDO cell models were established by Y.W, D.S., M. Silvestrov, C.R.A.R., R.S., J.L.R. and M.L. and drug sensitivity assays and corresponding analysis were performed by K.B under the leadership of C. Reinhard and J.A.V. Xenografts establishment and drug response data were provided by M.K and M.B. under the supervision of J. Hoffmann. Correlation of drug responses with genetic landscape were performed by M. Schütte and N.A.A. Identification of gene signatures of drug responses was done by T.R and C.J. Statistical analyses were performed by T.R. and M. Schütte. Data visualization was done by N.A.A., M. Schütte, T.R., V.A., H.J.W., C.J. and T.K. R.Y. was responsible for implementing and populating the web resource and the OncoTrack database. Additional contributions: C.W., T.K., R.H. performed additional annotation analysis. U.L, M.N.,

D.W., P.G.-C., C. Reinhard and B.L. contributed to data interpretation and provided conceptual advice. H.L. conceptualized the project and contributed to statistical evaluation and interpretation of the data. D.H. and H.L. jointly coordinated the project. M.L.Y. wrote the manuscript, coordinated and supervised the data analysis. The manuscript was communicated to all authors and all had the opportunity to comment on, and approved the paper.

**Additional information**

**Competing financial interests:** Several of the authors are employees of the following pharmaceutical companies: Eli Lilly (K.B., J.V., C. Reinhard), Merck KGaA (D.W.), Boehringer Ingelheim (P.G.-C.), and Bayer-Pharma (D.H., M.L.). Other authors are employees of Alacris Theranostics (B.L. (CEO), M. Schütte, R.Y., C.W., T.B., T.K.), founder of Alacris Theranostics (H.L.), founder and CEO of cpo (C.R.A.R.), employees or shareholders of cpo (Y.W., M. Silvestrov, R.S., J. Hoffmann), employee and/or shareholder of EPO (J. Hoffmann,M.K.). None of these companies influenced the interpretation of the data, or the data reported, or financially profit by the publication of the results. The remaining authors declare no competing financial interests.

Moritz Schütte[1,*], Thomas Risch[2,*], Nilofar Abdavi-Azar[2,*], Karsten Boehnke[3,*], Dirk Schumacher[4,5,*], Marlen Keil[6], Reha Yildirimman[1], Christine Jandrasits[2], Tatiana Borodina[1], Vyacheslav Amstislavskiy[2], Catherine L. Worth[2], Caroline Schweiger[7], Sandra Liebs[8], Martin Lange[9], Hans-Jörg Warnatz[2], Lee M. Butcher[10,11], James E. Barrett[10], Marc Sultan[2], Christoph Wierling[1], Nicole Golob-Schwarzl[7,12], Sigurd Lax[13], Stefan Uranitsch[14], Michael Becker[6], Yvonne Welte[4,15], Joseph Lewis Regan[9], Maxine Silvestrov[4,15], Inge Kehler[8], Alberto Fusi[8], Thomas Kessler[1], Ralf Herwig[16], Ulf Landegren[17], Dirk Wienke[18], Mats Nilsson[17,19], Juan A. Velasco[3], Pilar Garin-Chesa[20], Christoph Reinhard[21], Stephan Beck[10], Reinhold Schäfer[4,5], Christian R.A. Regenbrecht[4,15,**], David Henderson[22], Bodo Lange[1,**], Johannes Haybaeck[7,12,**], Ulrich Keilholz[8,**], Jens Hoffmann[6,**], Hans Lehrach[1,2,23] & Marie-Laure Yaspo[2,**]

[1] Alacris Theranostics GmbH, Fabeckstr. 60-62, D-14195 Berlin, Germany. [2] Max Planck Institute for Molecular Genetics, Department of Vertebrate Genomics/Otto Warburg Laboratory Gene Regulation and Systems Biology of cancer, Ihnestrasse 73, D-14195 Berlin, Germany. [3] Eli Lilly and Company, Lilly Research Laboratories, Quantitative Biology, Avda. de la Industria 30, Alcobendas, 28108 Madrid, Spain. [4] Charité–Universitätsmedizin Berlin, Institute of Pathology, Laboratory for Molecular Tumour Pathology, Charitéplatz 1, 10117 Berlin, Germany. [5] German Cancer Consortium (DKTK), German Cancer Research Center (DKFZ), Im Neuenheimer Feld 280, 69192 Heidelberg, Germany. [6] Experimental Pharmacology and Oncology Berlin-Buch GmbH (EPO), Robert-Roessle-Str. 10, 13125 Berlin, Germany. [7] Institute of Pathology, Medical University of Graz, Auenbruggerplatz 25, 8036 Graz, Austria. [8] Charité-Universitätsmedizin, Charitéplatz 1, 10117 Berlin, Germany. [9] Bayer Pharma AG, Müllerstraße 178, 13353 Berlin, Germany. [10] UCL Cancer Institute, University College London, London WC1E 6BT, UK. [11] Department of Surgery and Cancer, Imperial College London, London W12 0NN, UK. [12] Center for Biomarker Research in Medicine, Stiftingtalstrasse 5, 8010 Graz, Austria. [13] Department of Pathology, Hospital Graz Süd-West, Göstinger Straße 22, 8020 Graz, Austria. [14] Department of Surgery, Hospital Brothers of Charity Graz, Marschallgasse 12, 8020 Graz, Austria. [15] CPO–Cellular Phenomics& Oncology, Berlin-Buch GmbH, Robert-Rössle-Str. 10, 13125 Berlin, Germany. [16] Max Planck Institute for Molecular Genetics, Department of Computational Molecular Biology, Ihnestrasse 73, D-14195 Berlin, Germany. [17] Department of Immunology, Genetics and Pathology, SciLifeLab, Uppsala University, Box 815, SE-751 08 Uppsala, Sweden. [18] Merck KGaA, Frankfurter Str. 250, 64293 Darmstadt, Germany. [19] Science for Life Laboratory, Department of Biochemistry and Biophysics, Stockholm University, Tomtebodavägem 23A, Solna, Stockholm 17165, Sweden. [20] Boehringer Ingelheim RCV GmbH & Co KG, Dr. Boehringer-Gasse 5-11, A-1121 Wien, Austria. [21] Eli Lilly and Company, Lilly Research Laboratories, Oncology Translational Research, Lilly Corporate Center, Indianapolis, Indiana 46285, USA. [22] Bayer Pharma AG, Global External Innovation & Alliances, Müllerstraße 178, 13353 Berlin, Germany. [23] Dahlem Centre for Genome Research and Medical Systems Biology, Fabeckstr. 60-62, 14195 Berlin, Germany. * These authors contributed equally to this work. ** These authors jointly supervised this work.

