## [Peer Review File · Nature Communications]

Reviewers' comments:

Reviewer #1 (Remarks to the Author):

the authors have made considerable edits in response to the initial reviews and have clearly satisfied concerns with the original submission.

Reviewer #2 (Remarks to the Author):

The revised version of this manuscript represents a considerable improvement over the original submission. The finding that much of the "discordances" between the primary tumors and the PDX/PDOs are likely due to ITH within the original tumor, rather than genetic drift arising from repeated passaging, is particularly interesting.

While most of the key technical concerns have been addressed, the challenge that this study now faces is one of readability. There is so much data presented that the key conclusions sometimes become buried amidst a forest of interesting but perhaps lesser-impact findings. Having 10 figures for a paper is impressive, but ultimately may detract from the overall message. To maximize the impact of this work, it may be worthwhile editing the main text of the paper to a more manageable length, focusing on the most essential findings.

Reviewer #3 (Remarks to the Author):

Overview:

59 PDX and 46 PDO (with 19 overlapping) derived from 106 colorectal cancer patients. Samples characterised by exome sequencing and RNA-seq and the PDX and PDO models screened for viability effects against 11 and 13 drugs respectively and molecular markers of sensitivity/resistance assessed.

This study should be assessed against the backdrop of other recent larger PDX drug studies, including that of Bertotti (116 KRAS wild-type CRC PDX & cetuximab), Sellers (approx 1000 PDX of different tissue types & 62 drugs) and Caldas (83 breast explants & 108 compounds).

Main comments:

Section - Comparative molecular landscapes of the OT 1 cohort and derived tumor models

1. The main purpose of this section should be a comparison of the frequencies of driver mutations and recurrent copy number alterations (e.g. HNF4A, SKT4, SMAD4 etc) in TCGA cohorts (n>500) and this study's PDO and PDX models. Are they broadly similar? This section up until line 3, page 6 feels more like a mutation landscape paper. Rather, the authors should replace Figures 2b and c with simple scatter plots of the freq of driver mutations/cna in TCGA vs OT, TCGA vs PDX and TCGA vs PDO. I found Fig 2c not very clear. The plots in Fig 3a would be useful to have as supplementary figures for all 59 PDX and 46 PDO samples.

Section - Transcriptome landscapes of tumors and their derived models

1. The comparison of the expression of the 38 signatures between the PDX/PDO models and

Patient samples is well done and nicely demonstrates the lack of immune response, inflammatory and stromal components in the former.

2. In Fig 5b, it is stated that the changes in glucuronidation etc are likely a consequence of culture in matrigel - these are quite important changes but could be a consequence more generally of cell culture - is there any evidence from, for example, gene expression data of the 2D non-matrigel CCLE colorectal cell lines (either array based or RNA-seq) that these modules are not overexpressed?

3. I can find no reference for where the raw bam files etc have been deposited e.g. EGA - has this been done?

Figure 4a - the label 'OT patient' upper left side (and that shows the 3 main groups derived from the meta-analysis of the 38 signatures) does not make any sense - something more descriptive will be less confusing for the reader.

Section - Patterns of drug response in CRC-derived models

1. An important question is whether the pattern of PDO sensitivity seen in one lab is broadly similar to a different set of PDO from another - for example, almost all of the 13 compounds screened here in the PDO lines are included in the recent Clevers Cell paper of 20 colorectal organoids x 89 compounds and IC50/AUC values are available in the supp data.

2. A glaring omission from the 13 compounds screened in the PDO are Irinotecan (one of the 3 most commonly used agents in CRC alongside 5FU and Oxaliplatin) and Cetuximab. In previous reply the authors pointed out that Irinotecan needs to be liver metabolised - this is true but almost all cell culture labs used the active metabolite SN-38 for that reason. Additionally, given the later focus of the paper on Cetuximab and a signature of response in RAS wild-type samples, the statement previously that the compound was not suitable for their robotic platform needs explained. Many groups including the Clevers/Garnett have successfully screened Cetuximab using liquid-handling platforms and my own lab has also done this. If for whatever reason the authors are unable to do this, then they should screen these 19 organoids 'manually' in either 384- or 96-well formats. Using such data, could the Cetuximab signature in Fig 9e have also been derived using the PDO models? This is quite an important question given the authors stance that their PDX models are more 'similar to the human tumors' - this could be proven in the setting of Cetuximab with the authors access to the EPO, NV and KF gene expression datasets.

Section - Discussion

The authors state that "PDXs outperformed PDOs in recapitulating the CRC molecular groups" and "Compound screening in the 59 PDX models, more closely related to the human tumours..." - I'm not convinced that the data are sufficient to make such a strong statement - from Fig 4a, there are probably more PDX models where the authors were unable to assign to any of the 3 molecular groups and the numbers of discordant clonal mutations between Patient/PDX and Patient/PDO in Fig 3b look relatively similar. As noted above, this is where the Cetuximab dataset would be extremely valuable in the PDO models.

Minor amendments:

Page 3, line 24 - should also include NRAS based on JY Douilliard et al, NEJM 2013 and recent guidelines <http://www.jnccn.org/content/12/7/1028.full>

Page 15, line 21 - ? should be 'PDO'

Page 16, line 22 - include NRAS

Page 23, line 16 - is this really the best reference of Cetuximab efficacy in RAS wild-type tumors?

Reviewer #4 (Remarks to the Author):

The MS represents an extremely large piece of work and makes at least 3 important observations:

1. Most important: the identification of a novel 16 gene predictive classifier for cetuximab that was validated in 2 independent xenograft cohorts and 1 independent patient cohort. This classifier appears to be better than BRAF/KRAS status at predicting response to cetuximab and so is of potential clinical utility and interest to the CRC research community. Given the paper by Sclafani et al in JNCI in 2014, did the authors assess TP53 status in their analyses?
2. The "exquisite" sensitivity of the EML4-ALK fusion PDX to crizotinib is notable, but what is the frequency of this fusion in CRC? I take it this fusion was present in the equivalent primary tumour?
3. The relative lack of correlations between PDX and PDO models and even between the primary tumours and PDXs is an important observation and a cautionary lesson for the field.

Issues

1. Using a cut-off of 20% tumour content means that some of the primary samples used in the analyses will have had very little tumour content. As pointed out in the recent study by Dunne et al in Clinical Cancer Research 2016, sampling region is of major importance when developing molecular profiles based on transcriptional profiling due to the potential of stromal "contamination" of the tumour tissue to dominate the transcriptional signal. This may account for their relative lack of concordance with the CMS classification.
2. The authors have used a relatively large number of tumours and models, but is it sufficient to capture the complexity and heterogeneity of the disease? Did they perform power calculations to identify the number of models needed?
3. Is it not very surprising that some mutations were found in the models and not the primary tumour? If the tumours had been sequenced to a sufficient depth, it would be expected that all mutations found in the models would be picked up albeit potentially at low frequency? Were different regions of the primary molecularly profiled than were used for establishing PDX/PDO models?
4. The authors should comment on the relative value of the PDOs given their upregulation of drug resistant genes, lack of vasculature and the difficulty of treating with clinically relevant concentrations of drugs. What is the point of assessing angiogenesis inhibitors in these models?
5. Why did the authors use oxaliplatin alone when it is used exclusively in combination with 5FU? Was dialysed fetal calf serum (dFCS) used in the PDO studies into 5FU sensitivity? If not, the relevance of these experiments is questionable as dFCS is necessary to model the impact of 5FU on thymidylate synthase (TS) – in normal serum, the inhibition of TS by 5FU metabolite FdUMP can be overcome by the thymidylate salvage pathway mediated by thymidine kinase. This could explain the lack of correlation between sibling PDO and PDX models for 5FU.
6. Numbers are clearly an issue (see point 2) as the authors were unable to generate classifiers for a number of the drugs assessed due to a lack of responders.

Question

The MSI and MSS tumours both have a wide range of mutations – do later stage MSI and MSS tumours tend to have more mutations?

Recommendation

The study is perhaps over-ambitious and is difficult to follow in sections. I would prefer the MS to focus on the Cetuximab/EGFR story, maybe including the other RTK inhibitors. I would drop the chemotherapy analyses, which are potentially flawed and lack the clinical relevance of combination therapies FOLFOX and FOLFIRI. The authors themselves admit that the irinotecan data is difficult to interpret because of the high conversion of irinotecan to SN-38 in the mouse liver.

Answers to the Reviewers' comments:

Reviewer #1 (Remarks to the Author):

the authors have made considerable edits in response to the initial reviews and have clearly satisfied concerns with the original submission.

Reviewer #2 (Remarks to the Author):

The revised version of this manuscript represents a considerable improvement over the original submission. The finding that much of the “discordances” between the primary tumors and the PDX/PDOs are likely due to ITH within the original tumor, rather than genetic drift arising from repeated passaging, is particularly interesting.

While most of the key technical concerns have been addressed, the challenge that this study now faces is one of readability. There is so much data presented that the key conclusions sometimes become buried amidst a forest of interesting but perhaps lesser-impact findings. Having 10 figures for a paper is impressive, but ultimately may detract from the overall message. To maximize the impact of this work, it may be worthwhile editing the main text of the paper to a more manageable length, focusing on the most essential findings.

We have edited and streamlined the text, which is now within the length recommended by the journal's policy.

Reviewer #3 (Remarks to the Author):

Overview:

59 PDX and 46 PDO (with 19 overlapping) derived from 106 colorectal cancer patients. Samples characterised by exome sequencing and RNA-seq and the PDX and PDO models screened for viability effects against 11 and 13 drugs respectively and molecular markers of sensitivity/resistance assessed.

This study should be assessed against the backdrop of other recent larger PDX drug studies, including that of Bertotti (116 KRAS wild-type CRC PDX & cetuximab), Sellers (approx 1000 PDX of different tissue types & 62 drugs) and Caldas (83 breast explants & 108 compounds) .

We are not certain what the referee means, whether this is about numbers or about referencing. We had already referred to the works of Gao/Sellers and Bertotti in our manuscript, and concerning Gao/Sellers, we obviously referred to them given that we used their PDXs a validation cohort for our classifier. However, we have now strengthened this comparison with more details in the revised paper.

Here, we focus on CRC. To the best of our knowledge, we present the largest drug panel tested in a large *in vivo* colon cancer PDX study, and our work also reports on the largest tested CRC organoid collection. The Gao/Sellers and Bertotti studies are smaller than ours, either in size or in the number of drugs tested. Besides, neither of those studies referred to CRC molecular groups, nor had matched cells nor matched donor tumors (except for 18 donors/116 PDX in Bertotti).

Here, we have treated 59 PDXs with 11 drugs and 35 organoids with 15 drugs. Importantly, our biobank is representative of all CRC molecular subtypes, focusing on compounds clinically relevant for CRC. For cetuximab, we also generated RNAseq data for a second cohort of 60 PDXs (EPO cohort). Altogether, we include in the paper essential molecular information for 119 PDXs treated with cetuximab.

Gao et al. reported 59 CRC PDXs. Of these only 43 PDXs or fewer were treated with a given drug, and this was only with 21 drugs (not 62 !), the other compounds were tested only on one single PDX, the usefulness of which is questionable. Figure 2b of Gao shows only 36 CRC PDX, mainly treated with inhibitors of the MAP Kinase pathway. Only 41/43 of the Gao PDXs treated with cetuximab had associated RNAseq data, whereas we provide RNAseq

results for 58 treated PDXs. For cetuximab, their data are comparable to ours and to the results seen in clinical trials, further highlighting the relevance of using PDX models in pre-clinical drug screens. Thus, we used the 36 cetuximab-treated PDXs from Gao as one of the validation cohorts for our drug response classifier.

Bertotti et al. have treated 116 KRAS wt PDX with cetuximab only, observing a response rate comparable to ours, however no other drugs were tested. Further, the Bertotti cohort interrogates only KRAS wt tumors, does not provide expression data and reports only targeted sequencing for 18 donor tumors.

Finally, it is not all about numbers. We and others strongly believe that the value of a cohort study lies in its associated informative molecular and functional data. In this respect, our pre-clinical biobank, besides being the largest, is also the first to make available deep molecular characterization of both the donor tumors and the derived models.

The Caldas paper on breast cancer only came out during the second review of our manuscript, so we could not have mentioned it before. We are now referring to it in our revised version, in the context of tumor heterogeneity.

Main comments:

Section - Comparative molecular landscapes of the OT 1 cohort and derived tumor models

1. *The main purpose of this section should be a comparison of the frequencies of driver mutations and recurrent copy number alterations (e.g. HNF4A, SKT4, SMAD4 etc) in TCGA cohorts (n>500) and this study's PDO and PDX models. Are they broadly similar? This section up until line 3, page 6 feels more like a mutation landscape paper. Rather, the authors should replace Figures 2b and c with simple scatter plots of the freq of driver mutations/cna in TCGA vs OT, TCGA vs PDX and TCGA vs PDO. I found Fig 2c not very clear.*

We followed the suggestion to include these graphs. We have generated scatter plots and dot plots, comparing the frequency of TCGA and Pan cancer mutations with OT donors tumors, PDO and PDXs This figure has been incorporated as Supplementary Fig. 3. We agree that Fig 2c was not very intuitive, and it has now been replaced by a new dot plot figure.

Nonetheless, in our opinion, the purpose of this section is much more than a sanity check as to whether or not the genomic profile of the OT cohort is similar to TCGA, and we would maintain Fig. 2b. There is also a lack of consensus among the referees. Fig. 2b was precisely requested by referee nr.4 in the first review. Perhaps Fig. 2b might “feel” like a landscape paper, but genomic data are typically displayed in such figures just as heatmaps are used for expression profiles. There are only a few sentences about the genomic profiles in the paper, focusing on the comparison with models, so this is not like a landscape paper.

Comparing only the frequency of driver mutations to TCGA is not very informative. Such broad comparisons are only good if one does not have at hand the matching patient cohort, or when only a few mutations have been tested. Unlike previously reported pre-clinical PDX collections, we carried out a comprehensive comparative molecular analysis of CRC patient tumors/xenografts/organoids, displayed together with the stages of the tumors. Fig. 2b shows that metastases do not display a different mutation profile, for instance.

Further, Fig. 2b displays the combination (or absence) of mutations as well as fusions, amplifications and deletions, in each individual tumor and which of these tumors was modeled in PDO or in PDX. Not every reader is familiar with the complex pattern of alterations in CRCs, and we believe that Fig. 2b, allows at a glance to capture which type of tumor has been modeled.

Fig. 2b can be read together with Fig. 2c, introducing the ITH.

The plots in Fig 3a would be useful to have as supplementary figures for all 59 PDX and 46 PDO samples.

We have now added the plots for all informative samples (n=36) in Supplementary Fig. 5, the limiting factor being the extent of diploid regions allowing us to perform these calculations.

We should like to point out, however, that the corresponding data pertaining to the Sciclone analysis were already available in Supplementary table 8).

Section - Transcriptome landscapes of tumors and their derived models

1. The comparison of the expression of the 38 signatures between the PDX/PDO models and Patient samples is well done and nicely demonstrates the lack of immune response, inflammatory and stromal components in the former.

2. In Fig 5b, it is stated that the changes in glucuronidation etc are likely a consequence of culture in matrigel - these are quite important changes but could be a consequence more generally of cell culture - is there any evidence from, for example, gene expression data of the 2D non-matrigel CCLE colorectal cell lines (either array based or RNA-seq) that these modules are not overexpressed?

We thank the reviewer for this suggestion. We have compared our data with the CCLE resource and added this information to Fig. 5c (revised version). Interestingly, the expression levels of *UGTs* display a much wider range in the CCLE cell lines than in the OncoTrack organoids. This could be due to the fact that the organoids are primary cultures whose expression profiles were determined at low passages, whereas the CCLE data integrate mostly immortalized cell lines.

In any case, the expression levels of *UGTs*, and in particular of *SCD*, are significantly higher in the organoids, which could be a consequence of the matrigel, or of the organoid intrinsic features, it is hard to say. We have amended the statement accordingly.

3. I can find no reference for where the raw bam files etc have been deposited e.g. EGA - has this been done?

We are surprised to read this comment since the EGA accession number was provided on page 21 of the reviewed paper, in the section “Data and material access”

“The complete set of NGS data for patient tumors, xenografts and cell models is available in the European Genome-phenome Archive (EGA) of the EBI data repository under Accession number EGAS00001001752.”

In accordance with the terms of the patient informed consent and the IMI consortium agreement, the data are deposited in the restricted access section of the EGA and are therefore available on request.

Figure 4a - the label 'OT patient' upper left side (and that shows the 3 main groups derived from the meta-analysis of the 38 signatures) does not make any sense - something more descriptive will be less confusing for the reader.

We agree that the label was confusing. We have modified it to “OT patient tumor groups”.

Section - Patterns of drug response in CRC-derived models

1. An important question is whether the pattern of PDO sensitivity seen in one lab is broadly similar to a different set of PDO from another - for example, almost all of the 13 compounds screened here in the PDO lines are included in the recent Clevers Cell paper of 20 colorectal organoids x 89 compounds and IC50/AUC values are available in the supp data.

We compared the PDO sensitivity patterns for the seven compounds screened in common between our study and that of the recent Clevers paper, which tested 19 organoids. We have added this information in the revised paper .

It is important to note that the cell culture conditions as well as the analysis time points were different in the two studies. The OncoTrack study used

matrigel and serum free media, while Clevers maintained proliferative potential by using a human intestinal stem cell medium. Whereas Clevers et al used fixed time points for the drug treatment, we have determined doubling times for each respective PDO strain and adjusted the duration of treatment accordingly, as described in our materials & methods section.

In addition, different approaches taken in curve fitting may influence the interpretation of raw values. Clevers did an extrapolation of IC50s based on a few data points per drug, using a Markov Chain simulation as described in their material & methods section, while we employed the 4-parameter logistic CRC as described in Boehnke et al., 2016

(<http://jbx.sagepub.com/content/early/2016/05/26/1087057116650965.full>).

Both methods can be used to determine IC50 values and are equally popular. A direct comparison between the analysed datasets is difficult, but performing an in depth meta-analysis across diverse datasets generated with different parameters is out of the scope of our study. As more and more molecularly annotated datasets will become available, this will facilitate this type of analysis.

Nonetheless, we have compared the data for the seven drugs tested in common: AZD8931, Gefitinib, Irinotecan, Oxaliplatin, Sorafenib, *Cetuximab* and *5-FU*. To compare the data we convert the vdW $\ln(\text{IC}_{50})$ to regular IC50 and used only OT-PDOs with IC50 values below the assay limit, to compare only IC50's that were truly achieved.

Data are comparable for four drugs (AZD8931, Gefitinib, Irinotecan, and Sorafenib). The mean values of the OT IC50s are lower than those from the Clevers study, yet within a comparable micromolar range (see box plots below). For all OncoTrack response assays, the maximum concentration was 60 μM , besides cetuximab where a maximum concentration of 5 μM was used, as depicted with a blue dashed line. The maximal experimental concentrations for vdW/Clevers drug response assays are given within the table and indicated with a orange dashed line.

In contrast, the data for Oxaliplatin, Cetuximab and 5-FU (see below) were not comparable, given that the molarity range for IC50s values differed by orders of magnitude (see below).

We can explain these differences by the experimental setups of the two studies. We uniformly performed compound screening up to the limit of a maximum concentration of 60 μ M, in order to remain within a clinically meaningful range. This means, that for each tumor/drug individual assay for which an IC50 was not achieved by the 60 μ M maximum dose, we recorded that the sample was insensitive to that drug. In the data reported by vdW,

some of the reported IC50 values were generated by extrapolation, yielding theoretical IC50s that exceed the drug concentrations that were actually tested. We see comparable ranges of IC50s for those substances, where the actual IC50 was below 60 μ M, and thus reflecting objective responses.

The effects of compounds where an IC50 was reached only at much higher concentrations could not be rigorously compared.

Note: cetuximab data are discussed in the answer to the next referee's question.

Further, the OncoTrack donor tumors are well annotated in terms of tumor localisation, grade and stage and molecular types, to the best of our knowledge, those crucial annotations were not available for the cohort used in the Clevers paper.

2. A glaring omission from the 13 compounds screened in the PDO are Irinotecan (one of the 3 most commonly used agents in CRC alongside 5FU and Oxaliplatin) and Cetuximab.

We are now providing additional PDO data for cetuximab and irinotecan, although in our opinion these data do not add substantial information to our paper (see below).

In previous reply the authors pointed out that Irinotecan needs to be liver metabolised - this is true but almost all cell culture labs used the active metabolite SN-38 for that reason.

What we wanted to say in our reply to the first review is that it is not straightforward to draw conclusions about a pro-drug like CP-11 (irinotecan), which could undergo transformations by several routes.

In man, conversion of CPT-11 to SN-38 is achieved mainly by CES2 (human intestinal carboxylesterase 2), which is 60x more active than the related enzyme CES1 in activating CP-11. However, the overall SN-38 concentration is precisely determined by a balance between generation of SN-38 and oxidation/inactivation of CPT-11 by CYP3A4) and inactivation of SN-38 by

glucuronidation through UGT1A1/10 (Sanghani et al. 2004.
<http://dmd.aspetjournals.org/content/dmd/32/5/505.full.pdf>)

For CPT-11, the situation is rather complex since it is not only metabolized in the liver, but it is also converted locally in the bulk tumor. It turns out that the serum plasma concentration of the active compound SN-38 is not always correlated with positive response, which seem to be also determined by the local conversion of CP-11 into SN-38 by pancreatic tumors expressing CES2. (Capello et al.2015.
<http://jnci.oxfordjournals.org/content/107/8/djv132.full.pdf>).

There is ample recent literature reporting the use of irinotecan instead of SN-38 in cell cultures. (Weiswald et al., Br. J. Cancer 108: 1720-1731. 2013, Fan et al., Sci Rep. 6: 25062, 2016; Melatzki et al., PLoSOne 10: e0143194, 2016. Some groups even prefer using Irinotecan over SN-38 (Tardi et al. 2009. MolCancerTher 2009;8(8).

Irinotecan can be readily converted into SN38 as long as the treated cell lines express CES2 at reasonable levels. We are therefore convinced that testing irinotecan sensitivity in 3D-cultures is justified. Indeed, spheroid cultures of colon cancer lines were shown to produce SN-38 rapidly. The metabolite is localized preferably in the outer rim of spheroids, i.e. the region with mitotic activity (La Bonia et al., Proteomics 16, 1814-1821, 2016

Given that our organoids show high expression of CES2, (mean= 32 rpkm, median=18 rpkm) and virtually no expression of CYP3A4, thus avoiding to deplete the pool of CP-11), we treated 21 PDOs directly with irinotecan. The corresponding data are now incorporated in the revised manuscript.

Additionally, given the later focus of the paper on Cetuximab and a signature of response in RAS wild-type samples, the statement previously that the compound was not suitable for their robotic platform needs explained.

We provide more detailed explanations on the automatic compound dispensing process. Compounds/small molecules are normally dissolved in 100% DMSO. All dilution steps for the drug response assays are done in

DMSO/media solution. In the final organoid cell plate, all wells, including the positive control wells (MAX wells), contain 0.25% DMSO. Antibodies, however, are not handled in DMSO and therefore require a different liquid handling format and consequently, different positive/vehicle controls. Handling both DMSO and the antibody solvent would require operating two separate automated dilution platforms, challenging the concept of a high-throughput screening as presented here, which would ultimately affect the robustness of the assay. The implementation of complex three-dimensional organoid cultures as novel cell-based assay systems for drug screening requires automatic high-throughput platforms as well as comprehensive assay validation studies, as recently demonstrated in (Boehnke et al., J Biomol Screen. 2016 Oct;21(9):931--41).

Further, establishing a parallel platform for testing cetuximab in organoids would not contribute very useful data in our opinion, considering the limited efficiency of cetuximab *in vitro* in general (see discussion below).

Nonetheless, we have followed the reviewer's recommendations. We carried out a 'manual' screen for cetuximab on 19 OncoTrack organoids cultured in 384-well plate format. As expected, our data confirmed the marginal effect of cetuximab treatment in organoids (data presented below).

Many groups including the Clevers/Garnett have successfully screened Cetuximab using liquid-handling platforms and my own lab has also done this.

We are happy to address this interesting issue, although we are wondering what the referee precisely means by “successfully screened”. We are of a different opinion regarding the outcome of cetuximab treatment in cells.

In our view and as explained in more detail below, none of the CRC organoids used in the study by Clevers/Garnett (van de Wetering et al. 2015 Cell 161, 933–945) is responsive to cetuximab. Interpreting cetuximab sensitivity data requires a careful consideration of the following points:

1) Whether the range of cetuximab concentrations used in a given assay may objectively qualify a cellular response as significant or relevant with regard to the physiological/clinical conditions.

2) A detailed evaluation of the actual assay conditions used in the different *in vitro* setups, e.g. presence/absence of serum, treatment duration, drug concentration, etc. Drug effects reported for different culture conditions are not comparable, and even less so with 3D cell cultures.

The published data on the effect of cetuximab on cell viability in different cell culture systems is somewhat ambiguous. It was stipulated in the original publication describing cetuximab properties with regard to its preclinical development that C225 (cetuximab) achieved only modest growth inhibition *in vitro* (Prewett et al 1998), confirming earlier data from Goldstein et al., 1995; Ciardiello et al., 1996; Prewett et al., 1998). This trend is evident in more recent large-scale drug screenings. In the large CRC panel reported by Medico et al. (2015), 150 CRC cell lines were treated with cetuximab (10 micrograms/ml in serum-free medium (non-physiologic) for 4-5 days; not corrected for doubling times). Most KRAS/RAF wt cell lines (80% precisely) were unresponsive to cetuximab, even under the stringent non physiological conditions. Only 8 KRAS/RAF wt cell lines displayed RTK overexpression, which might explain resistance, but most of the wt samples were indeed insensitive to cetuximab.

In our internal unpublished data treating various CRC cell lines with cetuximab (2D, run in 10% serum; two doubling times), we never reached 50% inhibition. Even for the NCI-H-508 cell line, reported as sensitive to cetuximab in the Medico study (however under serum free conditions), we only reached 25% max inhibition under 10% serum conditions (this illustrates the influence of the serum on cetuximab response).

Considering the CRC organoid biobank reported by the Clevers group mentioned by the referee, which indeed represents the system most comparable to ours, we are somewhat perplexed by the reviewer's claim that cetuximab was successfully used in this platform. This is clearly not the case.

The Clevers's data on organoids shows that the % of inhibition reached with cetuximab is very low, precluding the calculation of reliable IC50s. Factually, IC50 is never reached in their study, simply, because the maximum inhibition rarely exceeds 20-30%, which means inactivity of the compound. See Figure S7 of the Clever's paper.

Figure S7

Further, although figure 7 of this publication is puzzling, it indicates no cetuximab activity at biologically relevant concentrations.

Figure 7

Our understanding is that this curve has been generated by data simulation, with the cetuximab concentrations tested in assays being in a completely different range than the estimated IC₅₀ (see the red box superimposed on the figure above). This can also be seen in the scatter plot of Figure 7c in the van de Wetering paper. Here, the estimated IC₅₀ values for cetuximab are in a range of ca. 20 μM to 1 mM, and thus orders of magnitude above the concentrations at which the antibody is normally used in cell culture experiments making these IC₅₀ values biologically and clinically meaningless by all objective measures.

Commonly, the highest concentration used for cetuximab in preclinical studies is in the range of 10-100 micrograms/ml, corresponding to 66 to 660 nM (Ashraf et al., 2012, PNAS 109, 21046-21051; Medico et al. 2015, Nature Communications, 7002). At these concentrations cetuximab completely blocks ligand induced EGFR phosphorylation (Ashraf et al., 2012, PNAS 109, 21046-21051). Any cellular effects observed only at micromolar cetuximab concentrations and above are unrelated to cetuximab-mediated EGFR blockade, and thus not meaningful biologically. Given that cetuximab is only available as 5 mg/ml (33 micromolar) solution, it is unlikely that anyone would conduct experiments with cetuximab concentrations higher than that.

The data in the Van de Wetering paper clearly show that cetuximab treatment does not trigger strong anti-proliferative effects in the 3D organoids used for that study.

If for whatever reason the authors are unable to do this, then they should screen these 19 organoids 'manually' in either 384- or 96-well formats.

We have screened 19 PDOs (7 wild-type for *KRAS/NRAS/BRAF*) manually in 384-wells format with a maximum cetuximab concentration of 5 micrograms/ml (32,89 nM). With the exception of one moderate responder sample (162_T_CELL, WT for *NRAS, KRAS, BRAF*), we did not detect objective response to cetuximab in our organoid biobank. For the two wild-type sibling pairs of xenograft and organoid, cetuximab sensitivity was higher in the *in vivo* setting. We have included these data in the Supplementary Fig. 15 and Supplementary Table 14 in the revised manuscript,. Our data are in line with the observations of the Clevers lab, who show in their validation screen only one sample with inhibition >50% at the highest concentration point tested.

In contrast to the Clevers paper, we did not extrapolate the dose-response curve fitting beyond the tested concentration range in order to fit the curves to 100% of growth inhibition to calculate IC50 values. Instead, we focused our analysis of the response data within a given clinically compatible concentration range. In line with the findings of the Clevers paper, we observed sensitivity upon treatment with the EGFR inhibitors AZD8931 and Gefitinib compared to Cetuximab treatment. In addition, we show response data for Afatinib. Thus, we strongly believe that we have appropriately addressed the effect of EGFR blockade in our organoid panel with three EGFR inhibitors.

Using such data, could the Cetuximab signature in Fig 9e have also been derived using the PDO models? This is quite an important question given the authors stance that their PDX models are more 'similar to the human tumors' -

this could be proven in the setting of Cetuximab with the authors access to the EPO, NV and KF gene expression datasets.

The answer to this relevant question is simply no. Owing to the very marginal effect of cetuximab in organoids, it is impossible to derive a predictive signature from the cell data. Further, as we show for the small compound inhibitors of the EGFR pathway, the biological differences reflected by the expression profiles of PDXs and PDOs, and the plasticity of the cultured cells (Fig 5 of the revised version), lead to different signatures associated with drug response in both model systems.

We already wrote in the previous version of the paper that the signatures for EGFR blockade response were different in cells and xenografts, due to their inherent biological differences. We tried to make it more clear in the revised text.

This illustrates one asset of our study, comparing directly the molecular patterns and drug responses in a significant number of cells and xenografts derived from the same tumor sample.

Section - Discussion

The authors state that "PDXs outperformed PDOs in recapitulating the CRC molecular groups" and "Compound screening in the 59 PDX models, more closely related to the human tumours .." - I'm not convinced that the data are sufficient to make such a strong statement - from Fig 4a, there are probably more PDX models where the authors were unable to assign to any of the 3 molecular groups and the numbers of discordant clonal mutations between Patient/PDX and Patient/PDO in Fig 3b look relatively similar. As noted above, this is where the Cetuximab dataset would be extremely valuable in the PDO models.

Actually, we could assign all PDXs and all PDOs to molecular groups.

In terms of number/nature of discordant mutations, we already stated that PDO and PDX were similar. The statement above refers to their profiles of expression. However, we modulated the statement that PDXs outperformed PDOs in the revised version.

We drew the conclusion that the molecular groups appeared more stable in PDXs based on the fact that their corresponding NMFa or NMFb groups were more often similar to that of the original donor tumors than for the PDOs (e.g epithelial subtype in donors remained epithelial in PDX instead of stem type in PDOs for instance). Further, we showed that two fractions, one epithelial and one with stem cell features co-existed in the PDO cultures, giving the PDOs the possibility to switch between subtypes.

Further fig 5 (reviewed +revised version) shows convincingly that a number of processes are only present in the cells, and absent in the donor tumors and in the xenografts.

Coming back to the cetuximab response shown here and considering the dataset by the Clevers's group, we can safely conclude that the cetuximab treatment is only interpretable in the xenografts, corroborating previous statements.

Minor amendments:

Page 3, line 24 - should also include NRAS based on JY Douilliard et al, NEJM 2013 and recent guidelines

<http://www.jnccn.org/content/12/7/1028.full>

This reference is incorporated in the revised version.

Page 15, line 21 - ? should be 'PDO'

We thank the reviewer for spotting this and we have corrected the typo.

Page 16, line 22 - include NRAS

This is now included.

Page 23, line 16 - is this really the best reference of Cetuximab efficacy in RAS wild-type tumors?

We thank the reviewer for spotting this error in the referencing. The appropriate references are Van Cutsem et al. Journal of Clinical oncology Vol. 33, Nb.7 , 2015 and Bokemeyer et al. European Journal of Cancer (2012) 48, 1466– 1475.

The manuscript has been edited accordingly.

Reviewer #4 (Remarks to the Author):

The MS represents an extremely large piece of work and makes at least 3 important observations:

1. Most important: the identification of a novel 16 gene predictive classifier for cetuximab that was validated in 2 independent xenograft cohorts and 1 independent patient cohort. This classifier appears to be better than BRAF/KRAS status at predicting response to cetuximab and so is of potential clinical utility and interest to the CRC research community. Given the paper by Sclafani et al in JNCI in 2014, did the authors assess TP53 status in their analyses?

In our study, the TP53 mutation status of the xenograft tumors did not correlate with the response profile to cetuximab. Of the 14 PDXs displaying sensitivity to cetuximab (criteria T/C <25) 12 had TP53 mutations, whereas 22/34 of the resistant ones were mutated for TP53. We do not see any correlation between wild-type TP53 and cetuximab response (p-value Fisher's exact test=0.18).

The Sclafani study compared CAPOX monotherapy versus CAPOX+cetuximab. They showed that TP53 wild-type patients treated with CAPOX+cetuximab have a significant improvement in progression free survival and overall survival as compared to patients treated with monotherapy (p-value 0.02). In the paper, the author claim that chemotherapy triggers TP53 signaling leading to increased sensitivity to EGFR blockade. In OncoTrack, we could not observe such an effect because we were in a different context using cetuximab as monotherapy.

2. The “exquisite” sensitivity of the EML4-ALK fusion PDX to crizotinib is notable, but what is the frequency of this fusion in CRC? I take it this fusion was present in the equivalent primary tumour?

The frequency of ALK fusions in CRC is approximately 1%. In the study by Lin et al., EML4-ALK fusions were found in 2/83 samples. Stransky et al. 2014 report one SMEK2-ALK fusion in 91 in rectal cancers but none in 286 in colon adenocarcinomas. Medico et al 2015 detected one EML4-ALK fusion among 151 CRC cell lines.

Lin, Eva, et al. "Exon array profiling detects EML4-ALK fusion in breast, colorectal, and non-small cell lung cancers." *Molecular Cancer Research* 7.9 (2009): 1466-1476.

Stransky, Nicolas, et al. "The landscape of kinase fusions in cancer." *Nature Comms* 5 (2014).

Medico, Enzo, et al. "The molecular landscape of colorectal cancer cell lines unveils clinically actionable kinase targets." *Nature Comms* 6 (2015).

We have only a small paraffin block but no frozen sample for the corresponding patient, which was therefore not described in our patient cohort.

3. The relative lack of correlations between PDX and PDO models and even between the primary tumours and PDXs is an important observation and a cautionary lesson for the field.

Issues

1. *Using a cut-off of 20% tumour content means that some of the primary samples used in the analyses will have had very little tumour content. As pointed out in the recent study by Dunne et al in Clinical Cancer Research 2016, sampling region is of major importance when developing molecular profiles based on transcriptional profiling due to the potential of stromal “contamination” of the tumour tissue to dominate the transcriptional signal*

We understand the referee’s concern, but the comment is based on a misunderstanding. Indeed, as indicated in the methods, we used the cutoff tumor purity $\geq 40\%$ (and not 20%) in all gene expression analysis and comparison with CMS classification.

The lower cutoff of 20% tumor purity was used only for scoring the somatic mutations (Figure 2b), a cutoff used by most labs analyzing cancer genomes.

Given that we systematically identified the mutated expressed reads in our mutation scoring, we could perform “orthogonal” validation of the detected mutations.

However, for the transcriptome analysis, we used only tumors with purity $\geq 40\%$ (Method part: page 33 line 28; page 35 line 7). The CMS-comparison was also carried out using samples with a purity $\geq 40\%$. We made this information more visible in the main figures of the manuscript.

The stromal contribution and immune infiltration represent integral components of the CRC profile, this is why most of the tumors have a purity close to 50%, and thus are reflected in the reported profiling analysis, not only in our paper but also in all other studies including the CMS classification. For instance, the ECM/EMTgroup, the equivalent of CMS4, is for a large part driven by stromal signatures in both studies.

This may account for their relative lack of concordance with the CMS classification.

We do not see a real lack of concordance with the CMS classification. As shown in Fig. 4a and Supplementary Fig.4d (reviewed manuscript – supplementary Fig. 6d in the revised version), our classification of the bulk patient tumors demonstrates that our data are in pretty good concordance with the CMS labels.

The issue illustrated on Fig. 4a is the discordant group labels obtained between the RF and SSP algorithms of the CMS classification itself. The CMS classification is based on two different algorithms, aimed at providing the most probable label for a given sample, which is the closest “centroid” globally defining a tumor class.

Our classification aims at reflecting the complex and heterogeneous molecular profiles identified in CRC tumors, highlighting also the detailed features of the samples that are biologically fitting “between two groups”, because they express pathway components that belong to each of those two groups (Fig. 4a). Those tumors are typically more difficult to classify with the CMS system, getting discordant labels with the two CMS algorithms, thus inherently showing apparently less concordance between our classification and the CMS labels.

The models are doing less well with the CMS classification precisely because some of the groups (e.g. CMS 4) are largely driven by the stromal components absent in the models. In contrast, our method can classify these samples.

The CMS classification is useful but it was not designed for performing a direct detailed comparison of the transcriptome profiles between bulk tumor/organoid/xenografts.

2. The authors have used a relatively large number of tumours and models, but is it sufficient to capture the complexity and heterogeneity of the disease? Did they perform power calculations to identify the number of models needed?

We are not sure whether the reviewer refers to the heterogeneity observed across different tumors reflected by the different tumor classes, or to intra-tumor heterogeneity (ITH).

Regarding the heterogeneity between the different the CRC tumors types, we are confident that our collection of models capture this level of complexity, as illustrated in Fig. 4. We derived our organoids and xenografts from a cohort representing all three major CRC sub-types in approximately equal proportions.

If the question relates to ITH, objective detailed answers will require a completely different study focused on ITH that is out of the scope of our manuscript, and addressing 1) what is the extent of ITH in the different tumors of the different groups, 2) the impact of ITH on CRC pathogenesis, 3) how many tumors display ITH impacting tumor treatment and 4) modeling the effects of the different tumor genotypes within the bulk tumor.

This is a very interesting aspect, but that was not the initial aim of our study. For some samples, ITH became apparent when comparing in details the tumor-derived models.

We do not think that performing power calculations will provide a very meaningful answer in our paper, without knowing the exact parameters, which need to be taken into account. Again, this would be a different paper.

3. Is it not very surprising that some mutations were found in the models and not the primary tumour?

Given the ITH seen in CRC and the inherent sampling during model establishment, this is not surprising. Inevitably, the part of the bulk tumor that has been sequenced is a different one than that used for establishing the models. In view of the ITH (e.g. Big Bang model of CRC, Sottoriva et al.2015), it is expected that some parts of the bulk tumors might carry “private” mutations not seen in other parts of the same tumor.

Van de Wetering et al, (2015) observed also discordant mutations between the CRC donor tumors and their derived organoid models. In more detail, 27/679 mutations (4 %) of cancer-significant genes (based on the PanCancer analysis) were found discordant, including for instance mutations in *APC* (third hit to that gene), *SMAD4* and *POLE*.

This issue had generally not been made very obvious in other publications reporting CRC models, mostly because only a few mutations were tested such as in Cho et al. (Colorectal cancer patient–derived xenografted tumors maintain characteristic features of the original tumors. *Journal of Surgical Research* 187.2:2014) in which only exon 15 of *BRAF*, exon 2 of *KRAS*, exons 9 and 20 of *PIK3CA*, exons 5–8 of *TP53*, and exon 16 of *APC*" 502-509 were tested), or because no matched patient data was available to make a comparison (Gao et al. 2015)

If the tumours had been sequenced to a sufficient depth, it would be expected that all mutations found in the models would be picked up albeit potentially at low frequency?

We have already addressed this question by deep targeted sequencing in the reviewed manuscript (Supplementary table 3).

In Figure 3b we mentioned that mutations in *PIK3CA*, *APC*, *SMAD4*, *TP53*, *MTOR* and *EGFR* diverging between models and donor tumors have been additionally checked by deep targeted sequencing.

But even at this depth, when the heterogeneity is localized in regionally distant parts of the tumors, it will not be picked up, unless many different parts of the tumors have been sequenced, which would represent a significant piece of work.

Were different regions of the primary molecularly profiled than were used for establishing PDX/PDO models?

Yes, this is virtually unavoidable in this type of scheme. Similar schemes are used by other labs (Caldas et al. 2016, van de Wetering et al, 2015).

However, in the light of our data and others who recently reported on ITH, this might encourage improved scheme and changing views in the field for instance by using more models, as we wrote in our paper "challenging the view of the 1x1x1 scheme proposed in Gao et al. 2015 "

4. The authors should comment on the relative value of the PDOs given their upregulation of drug resistant genes, lack of vasculature and the difficulty of treating with clinically relevant concentrations of drugs. What is the point of assessing angiogenesis inhibitors in these models?

Our point here was to compare PDOs and PDXs derived from the same bulk tumor. Our study highlights heterogeneity and different biological behavior, providing a rich data resource for further studies. PDOs are still useful in providing a convenient model that can be used for targeted mechanistic studies.

We agree that it could be questionable to test cells with angiogenesis inhibitors, but this still makes sense because most of those inhibitors are multi-kinase inhibitors, such as regorafenib and sorafenib, which can trigger effects by inhibiting other signaling pathways.

5. Why did the authors use oxaliplatin alone when it is used exclusively in combination with 5FU?

We have clearly focused our study on correlations with single drug activity in order to identify potential biomarkers associated with the response to these treatments.

We are aware that oxaliplatin and 5FU are not used as single agent in clinical settings, it is however relevant to investigate whether there is any response to those single agents, and whether the pattern of response/resistance could be correlated with specific molecular features, before considering the combination.

The PDX were not tested with FOLFIRI and FOLFOX, since the mouse models have a low tolerance to cumulative toxicity, making tests of combination therapies very difficult.

Was dialysed fetal calf serum (dFCS) used in the PDO studies into 5FU sensitivity? If not, the relevance of these experiments is questionable as dFCS is necessary to model the impact of 5FU on thymidylate synthase (TS) – in normal serum, the inhibition of TS by 5FU metabolite FdUMP can be overcome by the thymidylate salvage pathway mediated by thymidine kinase. This could explain the lack of correlation between sibling PDO and PDX models for 5FU.

The 3D *in vitro* cultures were embedded in matrigel (growth factor reduced) and maintained in serum-free medium. The composition of matrigel is

complex, including growth factors and a number of other proteins and compounds.

According by the study by Hughes et al (Proteomics, Volume 10, No. 9 May 2010, 1886–1890), which analyzed in detail the matrigel composition by mass spectrometry, thymidine kinases were not detected, demonstrating that TKs are either absent or represent only a very minor component of the matrigel.

Of note, TK1 and TK2 are expressed at comparable levels in tumors, xenografts, and organoids.

6. Numbers are clearly an issue (see point 2) as the authors were unable to generate classifiers for a number of the drugs assessed due to a lack of responders.

We do not agree with this statement. Whereas it is (almost) always safe to say “the more the better”, we have demonstrated that the n is not a problem here and that the size of our cohort is fitting the goals set in this study.

We had already provided explanations in our response to the first review. Further, we are not performing a statistical study here, report the results of a biobank in precision pre-clinical settings, looking individual tumors.

We have generated a unique biobank of 59 PDXs (treated with 11 drugs) and 35 organoids (treated with 15 drugs) representative of all CRC molecular subtypes, with detailed molecular analysis at the level of individual tumors compared with their models. To the best of our knowledge, we present the largest drug panel tested in a large representative of CRC *in vivo* PDX study (59 in OT vs. 43 for Gao, and vs. the 116 PDX from Bertotti being only KRAS wild-type and only treated for cetuximab), and *in vitro* organoids (35 vs. 19 for Clevers) and the largest number of matching pairs of PDX/organoids.

Further, by generating molecular data on a previous EPO cohort, we have a total of 116 PDXs treated with cetuximab, and with associated RNAseq data, which is larger than any study published. The study by Bertotti does not provide this type of data and in wtKRAS biased

The number of OT models is sufficient to identify a signature. This is demonstrated by the fact that the cetuximab predictor could be validated in

independent cohorts. Further, we are able to correlate molecular groups to our drug response classifiers.

There is simply (and unfortunately) not always a clear gene expression pattern associated with the drug response, irrespective of the n . If we look at the BI compound tested in our study, for instance, which triggers very good response in a good number of models, we could not find a very good signature predictive of response, probably because there are several pathways involved, operating at a different level in the different models. Whenever there are multiple biological mechanisms leading e.g. to resistance, we can expect difficulties in identifying signatures.

How many samples are necessary is highly depending on the drug. For instance, it is not possible to identify a signature of response to cetuximab in cells, because of their poor response to that drug. Only a handful of cell lines among 150 tested responded to cetuximab in Medico et al. Here this is more a matter of lack of sensitivity to cetuximab in cell culture than a problem with the n .

There are of course additional types of molecular markers to be investigated in the future, making our well annotated biobank a very useful resource for the community.

Question

The MSI and MSS tumours both have a wide range of mutations – do later stage MSI and MSS tumours tend to have more mutations?

We do not see such as trend in our cohort (Fig. 2b and Supplementary Table 3, reviewed and revised version of the manuscript).

In our cohort, the 8 MSI samples and the one non-MSI hypermutated tumor are all stage I-II, so we therefore we cannot comment whether these tumors have more mutations at a later stage. There is no significant difference between stage I MSI and stage II MSI tumors in terms of number of mutations.

Regarding the MSS tumors we do not see a significant difference in the number of mutations between early (I+II) and late stage (III+IV) tumors (see boxplot below).

(Test used: Wilcoxon-Mann-Whitney-Test, p-values are shown in the figure)

Recommendation

The study is perhaps over-ambitious and is difficult to follow in sections. I would prefer the MS to focus on the Cetuximab/EGFR story, maybe including the other RTK inhibitors. I would drop the chemotherapy analyses, which are potentially flawed and lack the clinical relevance of combination therapies FOLFOX and FOLFIRI.

We have now streamlined the text in the revised version. In the first round of revision, referees 2 and 4 requested to include the chemotherapies (irinotecan, 5FU and oxaliplatin), this is why we have them now in the revised version.

Further, the PDX were not tested with FOLFIRI and FOLFOX, since the mice have a low tolerance to cumulative toxicity, making tests of combination therapies very difficult. In addition, it is quite likely, that differences in metabolism or transport of the two drugs in a combination therapy will considerably complicate the interpretation of the data. Further, it would also be unlikely, that the same ratio of drugs in the combination, which is used in patients, will result in an equivalent effect in the mouse.

The authors themselves admit that the irinotecan data is difficult to interpret because of the high conversion of irinotecan to SN-38 in the mouse liver.

See above. Other referees asked us explicitly to include irinotecan in PDOs.

REVIEWERS' COMMENTS:

Reviewer #3 (Remarks to the Author):

I am satisfied that the authors have addressed my main concern, which was the lack of Cetuximab and Irinotecan data in the Patient-derived organoids – they have now screened both compounds in these lines and have included that data in the manuscript. They have also carried out the requested comparison of their own drug sensitivity data with that of the same drugs screened against colorectal organoids in the Clevers lab.

Based on these edits and additional experiments I am happy to proceed with publication notwithstanding the views of the other reviewers.

Reviewer #4 (Remarks to the Author):

The authors have addressed all my major concerns and responded adequately to my other queries and questions. I think this revised MS is a strong and extensive piece of work that provides important information to the field and should be accepted for publication.